# A New Concentration Inequality for Sampling Without Replacement and Its Application for Transductive Learning

**Yingzhen Yang** [1]

## Abstract

We introduce a new tool, Transductive Local Complexity (TLC), to analyze the generalization performance of transductive learning methods and motivate new transductive learning algorithms. Our work extends the idea of the popular Local Rademacher Complexity (LRC) (Bartlett et al., 2005) to the transductive setting with considerable and novel changes compared to the analysis of typical LRC methods in the inductive setting. While LRC has been widely used as a powerful tool in the analysis of inductive models with sharp generalization bounds for classification and minimax rates for nonparametric regression, it remains an open problem whether a localized version of Rademacher complexity based tool can be designed and applied to transductive learning and gain sharp bound for transductive learning which is consistent with the inductive excess risk bound by (LRC) (Bartlett et al., 2005). We give a confirmative answer to this open problem by TLC. Similar to the development of LRC (Bartlett & Mendelson, 2003), we build TLC by first establishing a novel and sharp concentration inequality for supremum of empirical processes for the gap between test and training loss in the setting of sampling uniformly without replacement. Then a peeling strategy and a new surrogate variance operator are used to derive the following excess risk bound in the transductive setting, which is consistent with that of the classical LRC based excess risk bound in the inductive setting. As an application of TLC, we use the new TLC tool to analyze the Transductive Kernel Learning (TKL) model, and derive sharper excess risk bound than that by the current state-of-the-art (Tolstikhin et al., 2014). As a result of independent interest, the concentration inequality for the test-train process is used to derive a sharp concentration inequality for the general supremum of empirical process involving random variables in the setting of sampling uniformly without replacement, with comparison to current concentration inequalities.

[1]School of Computing and Augmented Intelligence, Arizona State University, Tempe, AZ 85281, USA. Correspondence to: Yingzhen Yang <yingzhen.yang@asu.edu>.

*Proceedings of the $42^{nd}$ International Conference on Machine Learning*, Vancouver, Canada. PMLR 267, 2025. Copyright 2025 by the author(s).

## 1. Introduction

We study transductive learning in this paper, where the learner has access to both labeled training data and unlabeled test data, and the task is to predict the labels of the test data. Obtaining a tight generalization bound for transductive learning is an important problem in statistical learning theory. Tools for inductive learning, such as Rademacher complexity and VC dimension, have been used for transductive learning, including empirical risk minimization, transductive regression, and transductive classification (Vapnik, 1982; 1998; Cortes & Mohri, 2006; El-Yaniv & Pechyony, 2009). On the other hand, it is important to employ localized version of Rademacher complexity, such as Local Rademacher Complexity (LRC) (Bartlett et al., 2005), to obtain sharper generation bound for transductive learning, such as (Tolstikhin et al., 2014).

The classical work (LRC) (Bartlett et al., 2005) presents the following sharp bound for the excess risk of empirical risk minimizer $\widehat{f}$ for inductive learning as follows: for every $x > 0$, with probability at least $1 - \exp(-x)$,

$$
\text{Excess Risk of } \widehat{f} \le \Theta \left\{ \text{Fixed Point of the Sub-Root} \right.
$$
$$
\left. \text{Functions for Certain Empirical Process} + \frac{x}{N} \right\}, \quad (1)
$$

where $\Theta$ only hides a constant factor, and $N$ is the size of the training data. Given the fact that LRC is capable of achieving various minimax rates for M-estimators in tasks such as nonparametric regression in the inductive regime, we propose to solve the following interesting and important question for LRC based transductive learning:

Can we have a sharp LRC based generalization bound for the excess risk of transductive learning as that for the inductive setting?

The most relevant result which addresses the above open problem, to the best of our knowledge, is presented in (Tolstikhin et al., 2014, Corollary 14), where the excess risk bound is given as the following inequality which happens with high probability:

$$\text{Excess Risk of } \widehat{f} \leq \Theta \left( \frac{n}{u} r_m^* + \frac{n}{m} r_u^* + \frac{1}{m} + \frac{1}{u} \right).$$
(2)

Here $r_m^*$ and $r_u^*$ are the fixed points of upper bounds for certain empirical processes, where $m, u$ are the size of training data and test data. It is remarked that the above bound may diverge due to the undesirable factors of $n/m$ and $n/u$ before the fixed points. With $m$ or $u$ grows in a much slower rate than $n$ with $n = u + m$, $n/m \cdot r_u^* + n/u \cdot r_m^*$ may not converge to 0. An example for the standard transductive kernel learning is given in Section 4. As a result, there is a remarkable difference between the current state-of-the-art excess bound (2) in the transductive setting and the excess risk bound (1) in the inductive setting, and the latter always converges to 0 under standard learning models. We note that the excess risk bound in (Tolstikhin et al., 2014, Corollary 13) still diverges when $m = o(\sqrt{n})$ or $u = o(\sqrt{n})$ as $m, u \to \infty$.

Our main result is the following sharp bound for the excess risk below, such that with the same high probability as the inductive bound (1),

$$\text{Excess Risk of } \widehat{f} \leq \Theta \big( r_u + r_m + r^*$$
$$+ \frac{1}{u} + \frac{1}{m} + \frac{x}{\min\{u, m\}} \big), (16) \text{ in Theorem } 3.6. \quad (3)$$

Here $r_m, r_u, r^*$ are the fixed points of upper bounds for certain empirical processes, which all converge to 0 with a fast rate as the case in the popular inductive learning models. As a result of the sharp excess risk bound (3), we give a confirmative answer to the above open problem.

### 1.1. Summary of Main Results

Our main results are summarized as follows. This summary also features a high-level description of the ideas we have developed to obtain the detailed technical results in Section 3.

First, we present the first sharp bound for the excess risk of empirical minimizer for transductive learning using local complexities based method inspired by LRC (Bartlett et al., 2005), and such bound (3) is consistent with existing sharp bound for excess risk bound (1) for inductive

learning. Two novel technical elements are proposed to establish such sharp bound: (1) Transductive Local Complexity (TLC), which renders particularly sharp bound for transductive learning using the peeling strategy on the function class with a new surrogate variance operator; (2) a novel and sharp concentration inequality for the bound for the supremum of the empirical loss which is the difference between the test loss and the training loss, that is, $\sup_{h \in \mathcal{H}} (\mathcal{U}_h^u - \mathcal{L}_h^m)$, where $\mathcal{H}$ is a function class, $\mathcal{U}_h^u, \mathcal{L}_h^m$ are the test loss and the training loss associated with the predictor $h \in \mathcal{H}$. We refer to such empirical process as the test-train process in the sequel. It is remarked that the existing local complexity based transductive learning method (Tolstikhin et al., 2014) is based on the bound for the supremum of the empirical process which is the difference between the training or test loss and the population loss, that is, $\sup_{h \in \mathcal{H}} \mathcal{U}_h^u - \mathcal{L}_n(h)$ or $\sup_{h \in \mathcal{H}} \mathcal{L}_h^m - \mathcal{L}_n(h)$, where $\mathcal{L}_n(h)$ is the average loss of $h$ on the entire data. Our novel concentration inequality for the test-train process presented in Theorem 3.1 is derived using new techniques based on a novel and interesting property of the test-train process involving random variables in the setting of sampling uniformly without replacement and using the exponential version of the Efron-Stein inequality ((Boucheron et al., 2003, Theorem 2)) twice to derive the variance of the test-train process. As an application of our sharp bound for excess risk for generic transductive learning, we derive a sharp excess risk bound for transductive kernel learning by Theorem 4.1 in Section 4, which is sharper than current state-of-the-art (Tolstikhin et al., 2014).

Second, as a result of independent interest, we derive a sharp concentration inequality for the general supremum of empirical process involving random variables (RVs) in the setting of sampling uniformly without replacement in Theorem 5.1 in Section 5. Our new concentration inequality is sharper than the two versions of the concentration inequality in (Tolstikhin et al., 2014), and this result is based on our new concentration inequality for the test-train process introduced above.

It is worthwhile to mention that concentration inequalities about sampling without replacement have been actively studied in the literature (Bardenet & Maillard, 2015; Tolstikhin, 2017), including those on the multislice which are based on the modified log-Sobolev inequalities (Sambale & Sinulis, 2022). Compared to (Tolstikhin, 2017), our bound in Theorem 5.1 in Section 5 is sharper using a similar argument in Section 5. Furthermore, in contrast with our results, the supremum of empirical process involving sampling without replacement is not addressed in (Bardenet & Maillard, 2015).

## 1.2. Notations

We use bold letters for matrices and vectors, and regular lower letter for scalars throughout this paper. The bold letter with a single superscript indicates the corresponding column of a matrix, e.g. $\mathbf{A}_i$ is the $i$-th column of matrix $\mathbf{A}$, and the bold letter with subscripts indicates the corresponding element of a matrix or vector. We put an arrow on top of a letter with subscript if it denotes a vector, e.g., $\vec{\mathbf{x}}_i$ denotes the $i$-th training feature. We also use $\mathbf{Z}(i)$ to denote the $i$-th element of a vector $\mathbf{Z}$, and $\mathbf{Z}(i : j)$ denotes the vector formed by elemenets of $\mathbf{Z}$ with indices between $i$ and $j$ inclusively. Span $(() \mathbf{A})$ is the column space of matrix $\mathbf{A}$. $\|\cdot\|_F$ and $\|\cdot\|_p$ denote the Frobenius norm and the vector $\ell^p$-norm or the matrix $p$-norm. Var $[\cdot]$ denotes the variance of a random variable. $\mathbf{I}_n$ is a $n \times n$ identity matrix. $\mathbb{I}_{\{E\}}$ is an indicator function which takes the value of 1 if event $E$ happens, or 0 otherwise. The complement of a set $A$ is denoted by $\overline{A}$, and $|A|$ is the cardinality of the set $A$. tr $(\cdot)$ is the trace of a matrix. We denote the unit sphere in $d$-dimensional Euclidean space by $\mathbb{S}^{d-1} := \{\mathbf{x} \colon \mathbf{x} \in \mathbb{R}^d, \|\mathbf{x}\|_2 = 1\}$. Let $L^2(\mathcal{X}, \mu^{(P)})$ denote the space of square-integrable functions on $\mathbb{S}^{d-1}$ with probability measure $\mu^{(P)}$, and the inner product $\langle \cdot, \cdot \rangle_{\mu^{(P)}}$ and $\|\cdot\|_{\mu^{(P)}}^2$ are defined as $\langle f, g \rangle_{L^2} := \int_{\mathbb{S}^{d-1}} f(x)g(x)\mathrm{d}\mu^{(P)}(x)$ and $\|f\|_{L^2}^2 := \int_{\mathbb{S}^{d-1}} f^2(x)\mathrm{d}\mu^{(P)}(x) < \infty$. $\mathbb{P}_{\mathcal{A}}$ is the orthogonal projection onto a linear space $\mathcal{A}$, and $\mathcal{A}^\perp$ is the linear subspace orthogonal to $\mathcal{A}$. $\langle \cdot, \cdot \rangle_{\mathcal{H}}$ and $\|\cdot\|_{\mathcal{H}}$ denote the inner product and the norm in the Hilbert space $\mathcal{H}$. we write $a = \mathcal{O}(b)$ or $a \lesssim b$ if there exists a constant $C > 0$ such that $a \leq Cb$, $\tilde{\mathcal{O}}$ indicates there are specific requirements in the constants of the $\mathcal{O}$ notation. $a = o(b)$ and $a = w(b)$ indicates that $\lim |a/b| = 0$ and $\lim |a/b| = \infty$ respectively. $a \asymp b$ or $a = \Theta(b)$ denotes that there exists constants $c_1, c_2 > 0$ such that $c_1 b \leq a \leq c_2 b$. $\binom{m}{k}$ for $1 \leq k \leq m$ is the combinatory number of selecting $k$ different objects from $m$ objects. $\mathbb{R}^+$ is the set of all non-negative real numbers, and $\mathbb{N}$ is the set of all the natural numbers. We use the convention that $\sum_{i=p}^{q} = 0$ if $p > q$ or $q = 0$. $[m : n]$ denotes all the natural numbers between $m$ and $n$ inclusively, and we abbreviate $[1 : n]$ as $[n]$.

## 2. Problem Setup of Transductive Learning

We consider a set $\mathbf{S}_{m+u} := \left\{ (\vec{\mathbf{x}}_i, y_i) \right\}_{i=1}^{m+u}$, where $y_i$ is the label for the point $\vec{\mathbf{x}}_i$. Let $n = m + u$, $\left\{ \vec{\mathbf{x}}_i \right\}_{i=1}^{n} \subseteq \mathcal{X} \subseteq \mathbb{R}^d$, $\{y_i\}_{i=1}^{n} \subseteq \mathcal{Y} \subseteq \mathbb{R}$ where $\mathcal{X}, \mathcal{Y}$ are the input and output spaces. The learner is provided with the (unlabeled) full sample $\mathbf{X}_n := \left\{ \vec{\mathbf{x}}_i \right\}_{i=1}^{n}$. Under the standard setting of transductive learning (El-Yaniv & Pechyony, 2009; Tolstikhin et al., 2014), the training features $\mathbf{X}_m$ of size $m$ are sampled uniformly from $\mathbf{X}_n$ without replace-

ment, and the remaining features are the test features denoted by $\mathbf{X}_u = \mathbf{X}_n \setminus \mathbf{X}_m$. In the next paragraph we specificy the sampling process of $\mathbf{X}_u$ as a random subset of $\mathbf{X}_n$ of size $u$ sampled uniformly without replacement. Then it follows by symmetry that $\mathbf{X}_m$ are sampled uniformly from $\mathbf{X}_n$ without replacement.

Let $\mathbf{d} = [d_1, \ldots, d_u] \in \mathbb{N}^u$ be a random vector, and $\{d_i\}_{i=1}^{u}$ are $u$ independent random variables such that $d_i$ takes values in $[i : n]$ uniformly at random. Algorithm 1, which is adapted from (El-Yaniv & Pechyony, 2009) and deferred to the next subsection, specifies how to obtain $\mathbf{Z_d} = [\mathbf{Z_d}(1), \ldots, \mathbf{Z_d}(u)]^\top \in \mathbb{N}^u$ as the first $u$ elements of a uniformly distributed permutation of $[n]$, so that $\mathbf{Z_d}$ are the indices of $u$ test features sampled uniformly from $\mathbf{X}_n$ without replacement. Let $\mathbf{Z}$ be a vector, we use $\{\mathbf{Z}\}$ denote a set containing all the elements of the vector $\mathbf{Z}$ regardless of the order of these elements in $\mathbf{Z}$. Let $\overline{\mathbf{Z_d}} = [n] \setminus \{\mathbf{Z_d}\}$ be the indices not in $\{\mathbf{Z_d}\}$. It has been verified in (El-Yaniv & Pechyony, 2009) that the all the $u$ points in $\mathbf{X}_u := \left\{ \vec{\mathbf{x}}_i \right\}_{i \in \mathbf{Z_d}}$, which are selected by indices in $\{\mathbf{Z_d}\}$, are selected from $\mathbf{X}_n$ uniformly at random among all subsets of size $u$, and $\mathbf{X}_u$ serves as the test features. As a result, $\mathbf{X}_m = \mathbf{X}_n \setminus \mathbf{X}_u = \left\{ \vec{\mathbf{x}}_i \right\}_{i \in \overline{\mathbf{Z_d}}}$ are $m$ training features sampled uniformly from $\mathbf{X}_n$ without replacement. The training features together with their labels, $\{y_i\}_{i \in \overline{\mathbf{Z_d}}}$, are given to the learner as a training set. We denote the labeled training set by $\mathbf{S}_m := \left\{ \left( \vec{\mathbf{x}}_i, y_i \right) \right\}_{i \in \overline{\mathbf{Z_d}}}$. $\mathbf{X}_u$ is also called the test set. The learner's goal is to predict the labels of the test points in $\mathbf{X}_u$ based on $\mathbf{S}_m \bigcup \mathbf{X}_u$.

This paper studies the sharp generalization bounds of transductive learning algorithms. We assume that all the points in the full sample $\mathbf{X}_n$ are distinct. Given a prediction function $f$ defined on $\mathcal{X}$, we define the following loss functions. For simplicity of notations, we let $g(i) = g(\vec{\mathbf{x}}_i, y_i)$ or $g(i) = g(\vec{\mathbf{x}}_i)$ for a function $g$ defined on $\mathcal{X} \times \mathcal{Y}$ or $\mathcal{X}$. we write $\ell \circ f$ as $\ell_f$ and let $\ell_f(i) = \ell(f(\vec{\mathbf{x}}_i), y_i)$ be the loss on the $i$-th data point. Let $\mathcal{H}$ be a class of functions defined on $\mathcal{X} \times \mathcal{Y}$. For any set $\mathcal{A} \subseteq [n]$, we define $\mathcal{L}_h^{(m)}(\mathcal{A}) := 1/m \cdot \sum_{i \in \mathcal{A}} h(i)$ when $|\mathcal{A}| = m$, and $\mathcal{U}_h^{(u)}(\mathcal{A}) := \frac{1}{u} \sum_{i \in \mathcal{A}} h(i)$ when $|\mathcal{A}| = u$. The average loss and average squared loss associated with $h$ are defined as $\mathcal{L}_n(h) := \frac{1}{n} \sum_{i=1}^{n} h(i), T_n(h) := \frac{1}{n} \sum_{i=1}^{n} h^2(i)$. When $h = \ell_f$, $\mathcal{L}_h^{(m)}(\overline{\mathbf{Z_d}})$ and $\mathcal{U}_h^{(u)}(\mathbf{Z_d})$ are the training loss and the test loss of the prediction function $f$. We have $\mathbb{E}_{\mathbf{d}} \left[ \mathcal{U}_h^{(u)}(\mathbf{Z_d}) \right] = \mathbb{E}_{\mathbf{d}} \left[ \mathcal{L}_h^{(m)}(\overline{\mathbf{Z_d}}) \right] = \mathcal{L}_n(h)$.

### 2.1. Sampling Random Set Uniformly Without Replacement

The sampling strategy in (El-Yaniv & Pechyony, 2009) is adopted to sample $u$ points from the full sample $\mathbf{X}_n$ uniformly at random among all subsets of size $u$, which is described in Algorithm 1. Let $\mathbf{Z_d}$ be the vector returned by Algorithm 1. Then $\{\mathbf{Z_d}\}$ is the set of the indices of the test features, and $\overline{\mathbf{Z_d}}$ is the set of the indices of the training features.

---

**Algorithm 1** The RANDPERM Algorithm in (El-Yaniv & Pechyony, 2009), which obtains $\mathbf{Z_d} \in \mathbb{N}^u$ as the first $u$ elements of a uniformly distributed permutation of $[n]$ by sampling independent random variables $d_1, \ldots, d_u$.

---

1: $\mathbf{Z_d} \leftarrow \text{RANDPERM}(u)$
2: **input:** $u$
3: **initialize:** $\mathbf{I} = [n]$, $\mathbf{d}, \mathbf{Z_d} \in \mathbb{N}^u$ are initialized as zero vectors.
4: **for** $i = 1, \ldots, u$ **do**
  Sample $d_i$ uniformly from $[i : n]$.
  $\mathbf{d}(i) = d_i$, $\mathbf{Z_d}(i) = \mathbf{I}(d_i)$.
  Swap the values of $\mathbf{I}(i)$ and $\mathbf{I}(d_i)$.
5: **end for**
6: **return** $\mathbf{Z_d}$

---

### 2.2. Basic Definitions

We hereby define basic notations for Transductive Complexity. Let $\mathbf{d}' = [d_1', \ldots, d_u']$ be independent copies of $\mathbf{d}$, and $\mathbf{d}^{(i)} = [d_1, \ldots, d_{i-1}, d_i', d_{i+1}, \ldots, d_u]$. We define the supremum of the empirical process of the gap between the test loss and the training loss as

$$g(\mathbf{d}) := \sup_{h \in \mathcal{H}} \left( \mathcal{U}_h^{(u)}(\mathbf{Z_d}) - \mathcal{L}_h^{(m)}(\overline{\mathbf{Z_d}}) \right), \qquad (4)$$

where $\mathcal{U}_h^{(u)}(\mathbf{Z_d}) = \frac{1}{u} \sum_{i \in \{\mathbf{Z_d}\}} h(i)$ is the test loss, $\mathcal{L}_h^{(m)}(\overline{\mathbf{Z_d}}) = \frac{1}{m} \sum_{i \in \{\overline{\mathbf{Z_d}}\}} h(i)$ is the training loss. $g(\mathbf{d})$ is also referred to as the test-train process. We then define Rademacher variables and Transductive Complexity (TC), and then relate TC to the conventional inductive Rademacher complexity.

**Definition 2.1** (Rademacher Variables). Let $\{\sigma_i\}_{i=1}^n$ be $n$ i.i.d. random variables such that $\Pr[\sigma_i = 1] = \Pr[\sigma_i = -1] = \frac{1}{2}$, and they are defined as the Rademacher variables.

The Transductive Complexity is defined below.

**Definition 2.2** (Transductive Complexity). The four types of Transductive Complexity (TC) of a function class $\mathcal{H}$ are defined as

$$\mathfrak{R}_u^+(\mathcal{H}) := \mathbb{E}_{\mathbf{d}} \left[ \sup_{h \in \mathcal{H}} R_{u,\mathbf{d}}^+ h \right], \mathfrak{R}_u^-(\mathcal{H}) := \mathbb{E}_{\mathbf{d}} \left[ \sup_{h \in \mathcal{H}} R_{u,\mathbf{d}}^- h \right],$$

$$\mathfrak{R}_m^+(\mathcal{H}) := \mathbb{E}_{\mathbf{d}} \left[ \sup_{h \in \mathcal{H}} R_{m,\mathbf{d}}^+ h \right], \mathfrak{R}_m^-(\mathcal{H}) := \mathbb{E}_{\mathbf{d}} \left[ \sup_{h \in \mathcal{H}} R_{m,\mathbf{d}}^- h \right],$$

$$(5)$$

where $R_{u,\mathbf{d}}^+ h := 1/u \cdot \sum_{i=1}^{u} h(\mathbf{Z_d}(i)) - \mathcal{L}_n(h)$, $R_{m,\mathbf{d}}^+ h := 1/m \cdot \sum_{i=1}^{m} h(\overline{\mathbf{Z_d}}(i)) - \mathcal{L}_n(h)$, and $R_{u,\mathbf{d}}^- h := -R_{u,\mathbf{d}}^+ h$, $R_{m,\mathbf{d}}^- h := -R_{m,\mathbf{d}}^+ h$.

We remark that the proposed Transductive Complexity (TC) is fundamentally different from the transductive version of the Rademacher complexity in (El-Yaniv & Pechyony, 2009, Definition 1) in the sense that our TC is defined on the random training set or test set, while the counterpart in (El-Yaniv & Pechyony, 2009, Definition 1) operates on the entire full sample.

Let $\mathbf{Y}^{(u)} = \{Y_1, \ldots, Y_u\}$ with each $Y_i$ sampled uniformly and independently from $[n]$ with replacement for all $i \in [u]$. Similary, $\mathbf{Y}^{(m)} = \{Y_1, \ldots, Y_m\}$ with each $Y_i$ sampled uniformly and independently from $[n]$ with replacement for all $i \in [m]$. The following theorem relates the TC defined in Definition 2.2 to the usual inductive Rademacher complexity.

**Theorem 2.1.** Let $\boldsymbol{\sigma} = \{\sigma_i\}_{i=1}^{\max\{u,m\}}$ be iid Rademacher variables. Define $R_{\boldsymbol{\sigma}, \mathbf{Y}^{(u)}}^{(\text{ind})} h := \frac{1}{u} \sum_{i=1}^{u} \sigma_i h(Y_i)$ and $R_{\boldsymbol{\sigma}, \mathbf{Y}^{(m)}}^{(\text{ind})} h := \frac{1}{m} \sum_{i=1}^{m} \sigma_i h(Y_i)$. Then

$$\max \left\{ \mathfrak{R}_u^+(\mathcal{H}), \mathfrak{R}_u^-(\mathcal{H}) \right\} \leq 2\mathfrak{R}_u^{(\text{ind})}(\mathcal{H}),$$

$$\max \left\{ \mathfrak{R}_m^+(\mathcal{H}), \mathfrak{R}_m^-(\mathcal{H}) \right\} \leq 2\mathfrak{R}_m^{(\text{ind})}(\mathcal{H}), \qquad (6)$$

where $\mathfrak{R}_u^{(\text{ind})}(\mathcal{H}) := \mathbb{E}_{\mathbf{Y}^{(u)}, \boldsymbol{\sigma}} \left[ \sup_{h \in \mathcal{H}} R_{\boldsymbol{\sigma}, \mathbf{Y}^{(u)}}^{(\text{ind})} h \right]$, $\mathfrak{R}_m^{(\text{ind})}(\mathcal{H}) := \mathbb{E}_{\mathbf{Y}^{(m)}, \boldsymbol{\sigma}} \left[ \sup_{h \in \mathcal{H}} R_{\boldsymbol{\sigma}, \mathbf{Y}^{(m)}}^{(\text{ind})} h \right]$.

**Remark 2.2.** $\mathfrak{R}_u^{(\text{ind})}(\mathcal{H})$ and $\mathfrak{R}_m^{(\text{ind})}(\mathcal{H})$ are the Rademacher complexity in the inductive setting. It is remarked that (6) indicates that the established symmetrization inequality of inductive Rademacher complexity also holds for the transductive complexity defined in Definition 2.2. For simplicity of notations if no confusion arises, we also write $\mathfrak{R}_u^{(\text{ind})}(\mathcal{H}) = \mathbb{E} \left[ \sup_{h \in \mathcal{H}} R_{\boldsymbol{\sigma}, \mathbf{Y}^{(u)}}^{(\text{ind})} h \right]$ and $\mathfrak{R}_m^{(\text{ind})}(\mathcal{H}) = \mathbb{E} \left[ \sup_{h \in \mathcal{H}} R_{\boldsymbol{\sigma}, \mathbf{Y}^{(m)}}^{(\text{ind})} h \right]$.

We define the sub-root function below, which will be extensively used for deriving sharp bounds based on transductive local complexity.

**Definition 2.3** (Sub-root function,(Bartlett et al., 2005, Definition 3.1)). A function $\psi\colon [0,\infty) \to [0,\infty)$ is sub-root if it is nonnegative, nondecreasing and if $\frac{\psi(r)}{\sqrt{r}}$ is non-increasing for $r > 0$.

# 3. TLC Excess Risk Bound for Generic Transductive Learning

In this section, we first introduce our new concentration inequality for the test-train process as Theorem 3.1 in Section 3.1. We then apply Theorem 3.1 to obtain Theorem 3.2, which presents the bound for the test-train process involving the fixed points of certain sub-root functions as the upper bounds for the TC of localized function classes. Based on Theorem 3.2, the generalization bound and excess risk bound for generic transductive learning are presented in Theorem 3.5 and Theorem 3.6, respectively.

## 3.1. Concentration Inequality for the Test-Train Process

Let $\mathcal{H}$ be a class of functions defined on $\mathcal{X} \times \mathcal{Y}$ and for any $h \in \mathcal{H}$, $0 \leq |h(i)| \leq H_0$ for all $i \in [n]$ with a positive number $H_0$. For a technical reason we let $H_0 \geq 2\sqrt{2}$ throughout this paper, which is achieved by setting $H_0 = \max\{2\sqrt{2}, \max_{i \in [n]} |h(i)|\}$. Without special notes the function class $\mathcal{H}$ is separable in this paper. Given the function class $\mathcal{H}$, we define the function class $\mathcal{H}^2 \coloneqq \{h^2 \mid h \in \mathcal{H}\}$ as the "squared version" of $\mathcal{H}$. We then have the following concentration inequality for the test-train process $g(\mathbf{d})$. We consider two cases throughout this paper, that is, $m \gg u^2$ or $u \gg m^2$.

**Theorem 3.1** (Concentration Inequality for the Test-Train Process (4)). Assume that there is a positive number $r > 0$ such that $\sup_{h \in \mathcal{H}} T_n(h^2) \leq r$. Suppose that $m \gg u^2$ or $u \gg m^2$. Then for every $x > 0$, with probability at least $1 - \exp(-x) - (\min\{m,u\})^2 / \max\{m,u\}$ over $\mathbf{d}$,

$$g(\mathbf{d}) \leq \mathbb{E}_{\mathbf{d}}\left[g(\mathbf{d})\right] + 8\sqrt{\frac{5rx}{\min\{u,m\}}}$$

$$+ 2\sqrt{2} \inf_{\alpha > 0} \left(\frac{\mathfrak{R}^+_{\min\{u,m\}}(\mathcal{H}^2)}{\alpha} + \frac{2\alpha x}{\min\{u,m\}}\right) + \frac{8H_0^2 x}{\min\{u,m\}}.$$
(7)

Here $\mathfrak{R}^+_u(\cdot), \mathfrak{R}^+_m(\cdot)$ are the Transductive Complexity defined in (5), and $\mathcal{H}^2 = \{h^2 \mid h \in \mathcal{H}\}$.

**Key Innovations in the Proof of of Theorem 3.1.** Proof of Theorem 3.1 is deferred to Section B.2 of the appendix, and it is based on the a novel combinatorial property of the test-train process revealed in Lemma B.1 and Lemma B.2. Such property is used to derive the upper bound for the variance of $g(\mathbf{d})$, $V_+(g)$. Such upper bound also involves another empirical process for the class $\mathcal{H}^2$. The bound for

the empirical process for the $\mathcal{H}^2$ is derived with the exponential version of the Efron-Stein inequality ((Boucheron et al., 2003, Theorem 2)), and we use (Boucheron et al., 2003, Theorem 2) again along with the bound for $V_+(\cdot)$ to derive the sharp bound for $g(\mathbf{d})$.

## 3.2. The First Bound by Transductive Local Complexity

Using Theorem 3.1 and the peeling strategy in the proof of (Bartlett et al., 2005, Theorem 3.3), we have the following bound for the test-train process involving the fixed points of sub-root functions as the upper bounds for the TC of localized function classes.

**Theorem 3.2.** Suppose $K > 1$ is a fixed constant, and $\tilde{T}_n(h)\colon \mathcal{H} \to \mathbb{R}^+$ is a functional such that $T_n(h) \leq \tilde{T}_n(h)$ for all $h \in \mathcal{H}$. Let $\psi_u$ be a sub-root function and let $r_u$ be the fixed point of $\psi_u$. Let $\psi_m$ be another sub-root function and let $r_m$ be the fixed point of $\psi_m$. Assume that for all $r \geq r_u$,

$$\psi_u(r) \geq \max\left\{\mathbb{E}_{\mathbf{d}}\left[\sup_{h \in \mathcal{H}, \tilde{T}_n(h) \leq r} R^+_{u,\mathbf{d}} h\right],\right.$$
$$\left.\mathbb{E}_{\mathbf{d}}\left[\sup_{h \in \mathcal{H}, \tilde{T}_n(h) \leq r} R^+_{u,\mathbf{d}} h^2\right]\right\}, \quad (8)$$

and for all $r \geq r_m$,

$$\psi_m(r) \geq \max\left\{\mathbb{E}_{\mathbf{d}}\left[\sup_{h \in \mathcal{H}, \tilde{T}_n(h) \leq r} R^-_{m,\mathbf{d}} h\right],\right.$$
$$\left.\mathbb{E}_{\mathbf{d}}\left[\sup_{h \in \mathcal{H}, \tilde{T}_n(h) \leq r} R^+_{m,\mathbf{d}} h^2\right]\right\}. \quad (9)$$

Suppose that $m \gg u^2$ or $u \gg m^2$. Then for every $x > 0$, with probability at least $1 - \exp(-x) - (\min\{m,u\})^2 / \max\{m,u\}$ over $\mathbf{d}$, for every $h \in \mathcal{H}$,

$$\mathcal{U}_h^{(u)}(\mathbf{Z_d}) \leq \mathcal{L}_h^{(m)}(\overline{\mathbf{Z_d}}) + \frac{\tilde{T}_n(h)}{K} + c_0(r_u + r_m)$$
$$+ \frac{c_1 x}{\min\{m,u\}}, \quad (10)$$

where $c_0, c_1$ are absolute positive constants depending on $K$, and $c_1$ also depends on $H_0$.

**Remark 3.3.** $\tilde{T}_n(\cdot)$ is termed a surrogate variance operator, and it is an upper bound for the usual variance operator $T(\cdot)$. $\psi_u(r)$ is the sub-root upper bound for the TC for a localized function class, $\left\{h \in \mathcal{H}\colon \tilde{T}_n(h) \leq r\right\}$, where every function $h$ has its functional value $\tilde{T}_n(h)$ bounded by $r$. In this sense, $\psi_u(r)$ is the upper bound for the TC of a localized function class, so we attribute the results of Theorem 3.2 to Transductive Local Complexity (TLC). The same comments also apply to $\psi_m(r)$.

### 3.3. Sharp Excess Risk Bounds using Transductive Local Complexity (TLC) for Generic Transductive Learning

We apply Theorem 3.2 to the transductive learning task introduced in Section 2, and derive sharp bound for the excess risk. Suppose we have a function class $\mathcal{F}$ which contains all the prediction functions. We assume $0 \leq \ell_f(i) \leq L_0$ for all $f \in \mathcal{F}$ and all $i \in [n]$ throughout this paper, and $L_0 \geq 2\sqrt{2}$. Given $\mathbf{d}$, we define $\widehat{f}_{\mathbf{d},u} := \arg\min_{f \in \mathcal{F}} \mathcal{U}_{\ell_f}^{(u)}(\mathbf{Z_d})$ as the oracle predictor with minimum loss on the test set $\mathbf{X}_u$, and $\widehat{f}_{\mathbf{d},m} := \arg\min_{f \in \mathcal{F}} \mathcal{L}_{\ell_f}^{(m)}(\overline{\mathbf{Z_d}})$ as empirical minimizer, that is, the predictor with minimum loss on the training data $\mathbf{X}_m$. The excess risk of $f_{\mathbf{d},m}$ is defined by

$$\mathcal{E}(\widehat{f}_{\mathbf{d},m}) := \mathcal{U}_{\ell_{\widehat{f}_{\mathbf{d},m}}}^{(u)}(\mathbf{Z_d}) - \mathcal{U}_{\ell_{\widehat{f}_{\mathbf{d},u}}}^{(u)}(\mathbf{Z_d}). \quad (11)$$

Furthermore, we define the function class $\Delta_{\mathcal{F}} := \{h \colon h = \ell_{f_1} - \ell_{f_2}, f_1, f_2 \in \mathcal{F}\}$. For $h = \ell_{f_1} - \ell_{f_2} \in \Delta_{\mathcal{F}}$, we define in (12) a novel surrogate variance operator as a functional $\tilde{T}_n(h) \colon \Delta_{\mathcal{F}} \to \mathbb{R}+$ such that $T_n(h) \leq \tilde{T}_n(h)$. As a result, we can apply Theorem 3.2 to the functional class $\Delta_{\mathcal{F}}$ and obtain the following theorem, which states the upper bound for the test-train process with prediction functions as the difference of loss functions with two predictors from $\mathcal{F}$. The following assumption, which is the standard assumption adopted by existing local complexity based methods for both inductive and transductive learning (Bartlett et al., 2005; Tolstikhin et al., 2014) for performance guarantee of transductive learning with loss functions $\ell(\cdot, \cdot)$, is introduced below.

**Assumption 1** (Main Assumption). (1) There is a function $f_n^* \in \mathcal{F}$ such that $\ell_{f_n^*} = \inf_{f \in \mathcal{F}} \mathcal{L}_n(\ell_f)$.

(2) There is a constant $B$ such that for any $h \in \Delta_{\mathcal{F}}^*$, $T_n(h) \leq B\mathcal{L}_n(h)$, where $\Delta_{\mathcal{F}}^* := \{\ell_f - \ell_{f_n^*} \colon f \in \mathcal{F}\}$.

**Remark 3.4.** Assumption 1 is not restrictive, it is the standard assumption also used in (Tolstikhin et al., 2014). In addition, Assumption 1(2) holds if the loss function $\ell(\cdot, \cdot)$ is Lipschitz continuous in its first argument and a uniform convexity condition on $\ell$, for example, $\ell(y', y) = (y' - y)^2$.

Applying Theorem 3.2 to the function class $\Delta_{\mathcal{F}}$, we obtain the following theorem.

**Theorem 3.5.** Suppose that Assumption 1 holds, and $K > 1$ is a fixed constant. For $h = \ell_{f_1} - \ell_{f_2} \in \Delta_{\mathcal{F}}$ with $f_1, f_2 \in \mathcal{F}$, let

$$\tilde{T}_n(h) :=$$
$$\inf_{f_1, f_2 \in \mathcal{F} \colon \ell_{f_1} - \ell_{f_2} = h} 2B\mathcal{L}_n(\ell_{f_1} - \ell_{f_n^*}) + 2B\mathcal{L}_n(\ell_{f_2} - \ell_{f_n^*}). \quad (12)$$

Let $\psi_u$ be a sub-root function and $r_u$ is the fixed point of $\psi_u$. Let $\psi_m$ be another sub-root function and $r_m$ is the fixed point of $\psi_m$. Assume that for all $r \geq r_u$,

$$\psi_u(r) \geq \max \left\{ \mathbb{E}_{\mathbf{d}} \left[ \sup_{h \colon h \in \Delta_{\mathcal{F}}, \tilde{T}_n(h) \leq r} R_{u,\mathbf{d}}^+ h \right], \right.$$
$$\left. \mathbb{E}_{\mathbf{d}} \left[ \sup_{h \colon h \in \Delta_{\mathcal{F}}, \tilde{T}_n(h) \leq r} R_{u,\mathbf{d}}^+ h^2 \right] \right\}, \quad (13)$$

and for all $r \geq r_m$,

$$\psi_m(r) \geq \max \left\{ \mathbb{E}_{\mathbf{d}} \left[ \sup_{h \colon h \in \Delta_{\mathcal{F}}, \tilde{T}_n(h) \leq r} R_{m,\mathbf{d}}^- h \right], \right.$$
$$\left. \mathbb{E}_{\mathbf{d}} \left[ \sup_{h \colon h \in \Delta_{\mathcal{F}}, \tilde{T}_n(h) \leq r} R_{m,\mathbf{d}}^+ h^2 \right] \right\}. \quad (14)$$

Suppose that $m \gg u^2$ or $u \gg m^2$. Then for every $x > 0$, with probability at least $1 - \exp(-x) - (\min\{m, u\})^2 / \max\{m, u\}$ over $\mathbf{d}$, for every $h \in \Delta_{\mathcal{F}}$,

$$\mathcal{U}_h^{(u)}(\mathbf{Z_d}) \leq \mathcal{L}_h^{(m)}(\overline{\mathbf{Z_d}}) + \frac{2B}{K}\mathcal{L}_n(\ell_{f_1} - \ell_{f_n^*})$$
$$+ \frac{2B}{K}\mathcal{L}_n(\ell_{f_2} - \ell_{f_n^*}) + c_0(r_u + r_m) + \frac{c_1 x}{\min\{m, u\}}, \quad (15)$$

where $c_0, c_1$ are absolute positive constants depending on $K$, and $c_1$ also depends on $L_0$.

Combining Theorem 3.5 and Theorem B.11 in the appendix, we have the following excess risk bound for the empirical minimizer $\widehat{f}_{\mathbf{d},m}$.

**Theorem 3.6.** Suppose that Assumption 1 holds and $m \gg u^2$ or $u \gg m^2$, $K > 1$ is a fixed constant. Then for every $x > 0$, with probability at least $1 - 3\exp(-x) - 3(\min\{m, u\})^2 / \max\{m, u\}$ over $\mathbf{d}$, the excess risk $\mathcal{E}(\widehat{f}_{\mathbf{d},m})$ satisfies

$$\mathcal{E}(\widehat{f}_{\mathbf{d},m}) \leq c_0(r_u + r_m) + \frac{4Bc_2 r^*}{K} + \frac{c_3 x}{\min\{m, u\}}, \quad (16)$$

where $r^*$ is specified by Theorem B.11 in the appendix, $c_3 = c_1 + 4Bc_2/K$ is a positive constant, $c_0, c_1, c_2$ are the positive constants in Theorem 3.5 and Theorem B.11 in the appendix.

*Proof.* (16) follows by plugging the upper bounds (75) for $\mathcal{L}_n(\ell_{\widehat{f}_{\mathbf{d},u}} - \ell_{f_n^*})$ and $\mathcal{L}_n(\ell_{\widehat{f}_{\mathbf{d},m}} - \ell_{f_n^*})$ in Theorem B.11 to (15) in Theorem 3.5. $\square$

We remark that (16) is consistent with the sharp bound for excess risk for inductive learning (1), and there are only

constant factors on the fixed points $r_u$ and $r_m$. In contrast, the existing local complexity based excess risk bound (2) for transductive learning involves undesirable factors $n/u$ and $n/m$, which can make the bound (2) diverge under standard learning models, such as the Transductive Kernel Learning (TKL). In the next section, we apply the TLC based sharp excess risk bound (16) to TKL.

## 4. TLC Excess Risk Bound for Transductive Kernel Learning

We apply the results in Section 3.3 to obtain the sharper risk bound for transductive kernel learning in Theorem 4.1 than that in the current state-of-the-art (Tolstikhin et al., 2014).

**Background in RKHS and Kernel Learning.** Let $\mathcal{H}_K$ be the Reproducing Kernel Hilbert Space (RKHS) associated with $K$, where $K \colon \mathcal{X} \times \mathcal{X} \to \mathbb{R}$ is a positive definite kernel defined on the compact set $\mathcal{X} \times \mathcal{X}$, and we assume $\mathcal{X}$ is compact in this section. Let $\mathcal{H}_{\mathbf{X}_n} := \left\{ \sum_{i=1}^n K(\cdot, \vec{\mathbf{x}}_i) \alpha_i \,\middle|\, \{\alpha_i\}_{i=1}^n \subseteq \mathbb{R} \right\}$ be the usual RKHS spanned by $\left\{ K(\cdot, \vec{\mathbf{x}}_i) \right\}_{i=1}^n$ on the full sample $\mathbf{X}_n = \left\{ \vec{\mathbf{x}}_i \right\}_{i=1}^n$. Let the gram matrix of $K$ over the full sample be $\mathbf{K} \in \mathbb{R}^{n \times n}$, $\mathbf{K}_{ij} = K(\vec{\mathbf{x}}_i, \vec{\mathbf{x}}_j)$ for $i, j \in [n]$, and $\mathbf{K}_n := \frac{1}{n}\mathbf{K}$. Let $\widehat{\lambda}_1 \geq \widehat{\lambda}_2 \ldots \geq \widehat{\lambda}_n > 0$ be the eigenvalues of $\mathbf{K}_n$, and $\max_{\mathbf{x} \in \mathcal{X}} K(\mathbf{x}, \mathbf{x}) = \tau_0^2 < \infty$. We then have $\widehat{\lambda}_1 \leq \operatorname{tr}(\mathbf{K}_n) \leq \tau_0^2$. For a positive number $\mu$, define $\mathcal{H}_K(\mu) := \{ f \in \mathcal{H}_K \mid \|f\|_{\mathcal{H}} \leq \mu \}$, we consider the function class $\mathcal{H}_{\mathbf{X}_n}(\mu) := \mathcal{H}_{\mathbf{X}_n}(\mu) \bigcap \mathcal{H}_{\mathbf{X}_n}$ for TKL.

**Results.** The following assumption is standard when analyzing the LRC based excess risk bounds, which is also adopted in (Bartlett et al., 2005; Tolstikhin et al., 2014).

**Assumption 2.** (1) The loss function $\ell(\cdot, \cdot)$ is $L$-Lipschitz in its first argument, that is, $|\ell(f(\mathbf{x}), y) - \ell(f(\mathbf{x}'), y)| \leq L|f(\mathbf{x}) - f(\mathbf{x}')|$ for all $f \in \mathcal{F}$.

(2) There is a constant $B'$ such that for any $f \in \mathcal{F}$, $T_n(f - f_n^*) \leq B'\mathcal{L}_n(\ell_f - \ell_{f_n^*})$.

It can be verified that Assumption 2 implies Assumption 1 (2), so that the former is stronger than the latter. We now let the empirical minimizer $\widehat{f}_{\mathbf{d},m}$ and the oracle predictor $\widehat{f}_{\mathbf{d},u}$ be defined using the function class $\mathcal{F} = \mathcal{H}_{\mathbf{X}_n}(\mu)$. The following theorem states the sharp bound for the excess risk $\mathcal{E}(\widehat{f}_{\mathbf{d},m})$ for TKL based on Assumption 1 (1) and Assumption 2.

**Theorem 4.1.** Suppose that Assumption 1 (1) and Assumption 2 hold. Suppose $K$ is a positive definite kernel on $\mathcal{X} \times \mathcal{X}$. Suppose that for all $f \in \mathcal{H}_{\mathbf{X}_n}(\mu), 0 \leq \ell_f(i) \leq L_0$ for all $i \in [n]$, and $L_0 \geq 2\sqrt{2}$. Suppose that $m \gg u^2$ or

$u \gg m^2$. Then for every $x > 0$, with probability at least $1 - 3\exp(-x) - 3\left(\min\{m, u\}\right)^2 / \max\{m, u\}$ over $\mathbf{d}$, we have the excess risk bound

$$\mathcal{E}(\widehat{f}_{\mathbf{d},m}) \leq c_5 \left( \min_{0 \leq Q \leq n} r(u, m, Q) + \frac{x}{\min\{m, u\}} \right), \quad (17)$$

where

$$r(u, m, Q) := Q\left(\frac{1}{u} + \frac{1}{m}\right)$$
$$+ \left( \sqrt{\frac{\sum_{q=Q+1}^n \widehat{\lambda}_q}{u}} + \sqrt{\frac{\sum_{q=Q+1}^n \widehat{\lambda}_q}{m}} \right),$$

$c_5$ is an absolute positive constant depending on $B', L_0, L, \mu$.

**Comparison with current state-of-the-art.** We now compare our excess risk bound (17) for TKL to the following excess risk bound obtained by the current state-of-the-art method for TKL (Tolstikhin et al., 2014), which is also based on local complexity method for transductive learning. (Tolstikhin et al., 2014) shows that with high probability,

$$\mathcal{E}(\widehat{f}_{\mathbf{d},m}) \leq \Theta\left( \frac{n}{u} r_m^* + \frac{n}{m} r_u^* + \frac{1}{m} + \frac{1}{u} \right),$$
$$r_s^* \leq \Theta\left( \min_{0 \leq Q \leq s} \left( \frac{Q}{s} + \sqrt{\frac{\sum_{q=Q+1}^n \widehat{\lambda}_q}{s}} \right) \right), s = u \text{ or } m.$$
$$(18)$$

It is emphasized that both our excess risk bound (17) and (18) in (Tolstikhin et al., 2014) are derived under the same assumptions, Assumption 1 (1) and Assumption 2, and our result requires the additional assumption that $u \gg m^2$ or $m \gg u^2$. It can be observed that our bound (17) is free of the undesirable factors of $n/u$ and $n/m$ in (18). Moreover, it is well-known by standard results on the population and empirical Rademacher complexities (Bartlett et al., 2005; Mendelson, 2002) that when the full-sample $\mathbf{X}_n$ are sampled uniformly from the unit sphere with $\mathcal{X} = \mathbb{S}^{d-1}$ and the kernel $K$ is a dot-product kernel, then $\min_{0 \leq Q \leq n} \left( \frac{Q}{n} + \sqrt{\frac{\sum_{q=Q+1}^n \widehat{\lambda}_q}{n}} \right) \asymp n^{-2\alpha/(2\alpha+1)}$ with $\alpha > 1/2$. In this well studied case, the RHS of (18) diverges with $u = o\left(n^{1/(2\alpha+1)}\right)$ or $m = o\left(n^{1/(2\alpha+1)}\right)$, when $u, m \to \infty$. On the other hand, our excess risk bound (17) always converges to 0 as $u, m \to \infty$ because $r(u, n, Q) \leq \Theta\left(1/\sqrt{u} + 1/\sqrt{m}\right)$.

## 5. Concentration Inequality for Supremum of Empirical Process Involving RVs Sampled Uniformly Without Replacement

Suppose that $\mathcal{H}$ is a function class bounded by $H_0 \geq 2\sqrt{2}$ such that $\mathcal{L}_n(h) = 0$ for all $h \in \mathcal{H}$. Let $g_u(\mathbf{d}) := \sup_{h \in \mathcal{H}} \mathcal{U}_h^{(u)}(\mathbf{Z_d})$. The following theorem gives a sharp bound for such general empirical process $g_u(\mathbf{d})$. Such result follows from the concentration inequality for the test-train process in Theorem 3.1.

**Theorem 5.1.** Suppose $\sup_{h \in \mathcal{H}} T_n(h) \leq r$, and let $\mathcal{H}^2 = \left\{h^2 \colon h \in \mathcal{H}\right\}$. Suppose that $m \gg u^2$ or $u \gg m^2$. Then for every $x > 0$, with probability at least $1 - \exp(-x) - (\min\{m, u\})^2 / \max\{m, u\}$ over $\mathbf{d}$,

$$
g_u(\mathbf{d}) - \mathbb{E}_{\mathbf{d}}\left[g_u(\mathbf{d})\right] \lesssim \frac{m}{n}\left(\sqrt{\frac{rx}{\min\{u, m\}}}\right.
$$
$$
\left. + \inf_{\alpha > 0}\left(\frac{\mathfrak{R}_{\min\{u,m\}}^+(\mathcal{H}^2)}{\alpha} + \frac{\alpha x}{\min\{u, m\}}\right) + \frac{x}{\min\{u, m\}}\right).
$$
(19)

*Proof.* (19) follows immediately by (66) in Lemma B.10 and noting that $\mathcal{L}_n(h) = 0$. $\square$

We hereby compare the existing concentration inequalities for supremum of empirical process in (Tolstikhin et al., 2014). There are two versions of such inequalities in (Tolstikhin et al., 2014, Theroem 1), which are presented as follows. For the first version, with probability at least $1 - \exp(-t)$,

$$
g_u(\mathbf{d}) - \mathbb{E}_{\mathbf{d}}\left[g_u(\mathbf{d})\right] \leq 2\sqrt{\frac{2nrt}{u^2}}.
$$
(20)

For the second version in (Tolstikhin et al., 2014, Theroem 2), with probability at least $1 - \exp(-t)$,

$$
g_u(\mathbf{d}) - \mathbb{E}_{\mathbf{Y}^{(u)}}\left[\bar{g}_u(\mathbf{Y}^{(u))}\right]
$$
$$
\leq \sqrt{\frac{2(r + 2\mathbb{E}_{\mathbf{Y}^{(u)}}\left[\bar{g}_u(\mathbf{Y}^{(u))}\right])t}{u}} + \frac{t}{3},
$$
(21)

where $\bar{g}(\mathbf{Y}^{(u)}) := \sup_{h \in \mathcal{H}} \frac{1}{u} \cdot \sum_{i=1}^{u} h(Y_i)$ is the supremum of empirical process with iid random variables $\left\{bY^{(u)}\right\}$. Because we always expect the deviation between $g_u(\mathbf{d})$ and its expectation, the gap between $\mathbb{E}_{\mathbf{Y}^{(u)}}\left[\bar{g}_u(\mathbf{Y}^{(u))}\right]$ and $\mathbb{E}_{\mathbf{d}}\left[g_u(\mathbf{d})\right]$ is offered by (Tolstikhin et al., 2014, Theroem 3) as follows:

$$
0 \leq \mathbb{E}_{\mathbf{Y}^{(u)}}\left[\bar{g}_u(\mathbf{Y}^{(u))}\right] - \mathbb{E}_{\mathbf{d}}\left[g_u(\mathbf{d})\right] \leq \frac{2m^2}{n}.
$$

It follows from (21) and the above inequality that for the second version,

$$
g_u(\mathbf{d}) - \mathbb{E}_{\mathbf{d}}\left[g_u(\mathbf{d})\right]
$$
$$
\leq 2\sqrt{\frac{2(r + 2\mathbb{E}_{\mathbf{Y}^{(u)}}\left[\bar{g}_u(\mathbf{Y}^{(u))}\right])t}{u}} + \frac{t}{3} + \frac{2m^2}{n}.
$$
(22)

As a result, the RHS of (20) diverges when $u = o(\sqrt{n})$, and the RHS of (22) diverges when $m = w(\sqrt{n})$ as $u, m \to \infty$. In contrast, the RHS of our bound (19) converges to 0 under many standard learning models by noting that (1) $\mathfrak{R}_{\min\{u,m\}}^+(\mathcal{H}^2)$ can be bounded by the inductive Rademacher complexity of $\mathcal{H}^2$, $\mathfrak{R}_{\min\{u,m\}}^{(\text{ind})}(\mathcal{H}^2)$, using Theorem 2.1; (2) the inductive Rademacher complexity $\mathfrak{R}_{\min\{u,m\}}^{(\text{ind})}(\mathcal{H}^2)$ usually converges to 0 at a fast rate, such as $\mathcal{O}(\sqrt{1/\min\{u, m\}})$, for many standard learning models (Bartlett & Mendelson, 2003), when combined with the contraction property of the inductive Rademacher complexity (e.g., Theorem A.5).

## 6. Conclusion

We present Transductive Local Complexity (TLC) to derive sharp excess risk for transductive learning. TLC is based on our new concentration inequality for the supremum of empirical processes for the gap between the test and the training loss in the setting of sampling uniformly without replacement. Using a peeling strategy and a new surrogate variance operator, sharper excess risk bound, compared to the current state-of-the-art, for generic transductive learning with bounded loss function is derived. As an result of independent interest, a sharp concentration inequality for the general supremum of empirical process involving random variables in the setting of sampling uniformly without replacement is derived using the concentration inequality for the test-train process, with comparison to the current concentration inequalities.

## Acknowledgments

This work is supported by the 2023 Mayo Clinic and Arizona State University Alliance for Health Care Collaborative Research Seed Grant Program under Grant Award Number AWD00038846.

## Impact Statement

This paper presents work whose goal is to advance the theoretical understanding of sharp generalization capability of transductive learning.

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

We present the basic mathematical results required in our proofs in Section A, then present proofs of our results in this paper in Section B.

## A. Mathematical Tools

We introduce the basic concentration inequality which we used to develop the main results of this paper. Let $X_1, X_2, \ldots, X_n$ are independent random variables taking values in a measurable space $\mathcal{X}$, and let $X_1^n$ denote the vector of these $n$ random variables. Let $f \colon \mathcal{X}^n \to \mathbb{R}$ be some measurable function. We are concerned with concentration of the random variable $Z = f(X_1, X_2, \ldots, X_n)$. Let $X_1', X_2', \ldots, X_n'$ denote independent copies of $X_1, X_2, \ldots, X_n$, and we write

$$Z^{(i)} = f(X_1, \ldots, X_{i-1}, X_i', X_{i+1}, \ldots, X_n).$$

**Theorem A.1.** ((Boucheron et al., 2003, Theorem 2), the exponential version of the Efron-Stein inequality) For all $\theta > 0$ and $\lambda \in (0, 1/\theta)$,

$$\log \mathbb{E}\left[\exp\left(\lambda\left(Z - \mathbb{E}\left[Z\right]\right)\right)\right] \leq \frac{\lambda\theta}{1 - \lambda\theta} \log \mathbb{E}\left[\exp\left(\frac{\lambda V_+}{\theta}\right)\right]. \tag{23}$$

**Theorem A.2.** ((Boucheron et al., 2003, Theorem 5,Theorem 6)) Assume that there exist constants $a \geq 0$ and $b > 0$ such that

$$V_+(Z) := \mathbb{E}\left[\sum_{i=1}^n \left(Z - Z^{(i)}\right)^2 \mathbb{1}_{\left\{Z > Z^{(i)}\right\}} \mid X_1^n\right] \leq aZ + b.$$

Then for any $\lambda \in (0, 1/a)$,

$$\log \mathbb{E}\left[\exp\left(\lambda(Z - \mathbb{E}\left[Z\right])\right)\right] \leq \frac{\lambda^2}{1 - a\lambda}\left(a\mathbb{E}\left[Z\right] + b\right), \tag{24}$$

and for all $t > 0$,

$$\Pr\left[Z > \mathbb{E}\left[Z\right] + t\right] \leq \exp\left(\frac{-t^2}{4a\mathbb{E}\left[Z\right] + 4b + 2at}\right). \tag{25}$$

Moreover, if

$$V_-(Z) := \mathbb{E}\left[\sum_{i=1}^n \left(Z - Z^{(i)}\right)^2 \mathbb{1}_{\left\{Z < Z^{(i)}\right\}} \mid X_1^n\right] \leq v(Z)$$

holds for a nondecreasing function $v$. Then, for all $t > 0$,

$$\Pr\left[Z < \mathbb{E}\left[Z\right] - t\right] \leq \exp\left(\frac{-t^2}{4\mathbb{E}\left[v(Z)\right]}\right). \tag{26}$$

**Remark A.3.** While $a > 0$ in the original Theorem 5 of (Boucheron et al., 2003), one can use the same proof of this theorem to show that (25) holds for $a = 0$ with $b > 0$. $V_+(Z)$ defined in this theorem is the "upper variance" of the random variable as a function of independent random variables $X_1, X_2, \ldots, X_n$. In particular, $V_+(Z)$ measures the variance of $Z$ when $X_i$ is changed to another sample $X_i'$ for all $i \in [n]$.

**Proposition A.4.** (Logarithmic Sobolev inequality in (Boucheron et al., 2003, Proposition 10)), which is a variant proposed by (Massart, 2000)) For all $\lambda \in \mathbb{R}$,

$$\lambda\mathbb{E}\left[Z \exp(\lambda Z)\right] - \mathbb{E}\left[\exp(\lambda Z)\right] \log \mathbb{E}\left[Z \exp(\lambda Z)\right]$$

$$\leq \sum_{i=1}^n \mathbb{E}\left[\exp(\lambda Z)\psi\left(-\lambda(Z - Z^{(i)})\right) \mathbb{1}_{\left\{Z > Z^{(i)}\right\}}\right], \tag{27}$$

where $\psi(x) := x(e^x - 1)$.

**Theorem A.5** (Contraction Property of Inductive Rademacher Complexity (Ledoux & Talagrand, 1991)). Suppose $g$ is a Lipschitz continuous with $|g(x) - g(y)| \leq L|x - y|$. Then

$$\mathfrak{R}_u^{(\text{ind})}(g \circ \mathcal{H}) \leq L\mathfrak{R}_u^{(\text{ind})}(\mathcal{H}), \quad \mathfrak{R}_m^{(\text{ind})}(g \circ \mathcal{H}) \leq L\mathfrak{R}_m^{(\text{ind})}(\mathcal{H}).$$

**Theorem A.6** ((Hoeffding, 1963, Theorem 4), (Gross & Nesme, 2010, Section D)). Let $\{X_i\}_{i=1}^n$ and $\{Y_i\}_{i=1}^n$ be sampled from a population $\{c_i\}_{i=1}^N \subseteq \mathcal{X} \subseteq \mathbb{R}^d$ without replacement and with replacement respectively. Suppose $f$ is continuous and convex on $\mathcal{X}$, then

$$\mathbb{E}\left[f\left(\sum_{i=1}^n X_i\right)\right] \leq \mathbb{E}\left[f\left(\sum_{i=1}^n Y_i\right)\right]. \tag{28}$$

## B. Proofs

### B.1. Proof of Theorem 2.1

***Proof of Theorem 2.1.*** We prove the first bound in (6). We let $\mathbf{Y}^{(u)} = \{Y_1, \ldots, Y_u\}$ be $u$ independent random variables with each $Y_i$ for $i \in [u]$ sampled uniformly from $[n]$ with replacement. Let $\mathbf{Y}^{(u)'} = [Y_1', \ldots, Y_u']$ be independent copies of $\mathbf{Y}^{(u)}$, and $\boldsymbol{\sigma} = \{\sigma_i\}_{i=1}^{\max\{u,m\}}$ be iid Rademacher variables. Let $\mathcal{H}_0 = \left\{h_j^{(0)}\right\}_{j \geq 1}$ be a countable dense subset of $\mathcal{H}$ such that $\overline{\mathcal{H}_0} = \mathcal{H}$. We define $c_i = \left[h_j^{(0)}(i) - \mathcal{L}_n(h)\right]_{j \in [M]} \in \mathbb{R}^M$ for $i \in [n]$, and let $\{Q_i\}_{i \in [u]}$ and $\{Q_i'\}_{i \in [u]}$ be sampled from $\{c_i\}_{i \in [n]}$ without replacement and with replacement respectively. Then it follows from Theorem A.6 that $\mathbb{E}\left[f\left(\sum_{i=1}^u Q_i\right)\right] \leq \mathbb{E}\left[f\left(\sum_{i=1}^u Q_i'\right)\right]$, which means that

$$\mathbb{E}_{\mathbf{d}}\left[\max_{j \in [M]}\left(\mathcal{U}_{h_j^{(0)}}^{(u)}(\mathbf{Z}_{\mathbf{d}}) - \mathcal{L}_n(h_j^{(0)})\right)\right] \leq \mathbb{E}_{\mathbf{Y}^{(u)}}\left[\max_{j \in [M]}\left(\frac{1}{u}\sum_{i=1}^u h_j^{(0)}(Y_i) - \mathcal{L}_n(h_j^{(0)})\right)\right],$$

with $f(\mathbf{x}) = \max_{j \in [M]} \mathbf{x}_j$ being a convex function for $\mathbf{x} \in \mathbb{R}^M$, due to the fact that $\{\mathbf{Z}_{\mathbf{d}}\}$ is a random set of size $u$ sampled uniformly from $[n]$ without replacement. We note that both sequences $\left\{\max_{j \in [M]}\left(\mathcal{U}_{h_j^{(0)}}^{(u)}(\mathbf{Z}_{\mathbf{d}}) - \mathcal{L}_n(h_j^{(0)})\right)\right\}_{M \geq 1}$ and $\left\{\max_{j \in [M]}\left(\frac{1}{u}\sum_{i=1}^u h_j^{(0)}(Y_i) - \mathcal{L}_n(h_j^{(0)})\right)\right\}_{M \geq 1}$ are nondecreasing in terms of $M$. Letting $M \to \infty$, it then follows from Levi's monotone convergence theorem and the fact that the first element of both sequences are integrable that

$$\mathbb{E}_{\mathbf{d}}\left[\sup_{h \in \mathcal{H}_0}\left(\mathcal{U}_h^{(u)}(\mathbf{Z}_{\mathbf{d}}) - \mathcal{L}_n(h)\right)\right] \leq \mathbb{E}_{\mathbf{Y}^{(u)}}\left[\sup_{h \in \mathcal{H}_0}\left(\frac{1}{u}\sum_{i=1}^u h(Y_i) - \mathcal{L}_n(h)\right)\right].$$

Because $\mathcal{H}_0$ is dense in $\mathcal{H}$, we have

$$\mathbb{E}_{\mathbf{d}}\left[\sup_{h \in \mathcal{H}}\left(\mathcal{U}_h^{(u)}(\mathbf{Z}_{\mathbf{d}}) - \mathcal{L}_n(h)\right)\right] \leq \mathbb{E}_{\mathbf{Y}^{(u)}}\left[\sup_{h \in \mathcal{H}}\left(\frac{1}{u}\sum_{i=1}^u h(Y_i) - \mathcal{L}_n(h)\right)\right]. \tag{29}$$

As a result, we have

$$\begin{aligned}
\mathfrak{R}_u^+(\mathcal{H}) &= \mathbb{E}_{\mathbf{d}}\left[\sup_{h \in \mathcal{H}}\left(\mathcal{U}_h^{(u)}(\mathbf{Z}_{\mathbf{d}}) - \mathcal{L}_n(h)\right)\right] \\
&\overset{①}{\leq} \mathbb{E}_{\mathbf{Y}^{(u)}}\left[\sup_{h \in \mathcal{H}}\left(\frac{1}{u}\sum_{i=1}^u h(Y_i) - \mathcal{L}_n(h)\right)\right] \\
&\overset{②}{=} \mathbb{E}_{\mathbf{Y}^{(u)}}\left[\sup_{h \in \mathcal{H}}\left(\frac{1}{u}\sum_{i=1}^u h(Y_i) - \mathbb{E}_{\mathbf{Y}^{(u)'}}\left[\frac{1}{u}\sum_{i=1}^u h(Y_i')\right]\right)\right]
\end{aligned}$$

$$\stackrel{③}{\leq} \mathbb{E}_{\mathbf{Y}^{(u)}, \mathbf{Y}^{(u)'}} \left[ \frac{1}{u} \sup_{h \in \mathcal{H}} \left( \sum_{i=1}^{u} h(Y_i) - \sum_{i=1}^{u} h(Y_i') \right) \right]$$

$$\stackrel{④}{=} \mathbb{E}_{\mathbf{Y}^{(u)}, \mathbf{Y}^{(u)'}, \boldsymbol{\sigma}} \left[ \frac{1}{u} \sup_{h \in \mathcal{H}} \left( \sum_{i=1}^{u} \sigma_i \left( h(Y_i) - h(Y_i') \right) \right) \right]$$

$$\leq \mathbb{E}_{\mathbf{Y}^{(u)}, \boldsymbol{\sigma}} \left[ \frac{1}{u} \sup_{h \in \mathcal{H}} \sum_{i=1}^{u} \sigma_i h(Y_i) \right] + \mathbb{E}_{\mathbf{Y}^{(u)'}, \boldsymbol{\sigma}} \left[ \frac{1}{u} \sup_{h \in \mathcal{H}} \sum_{i=1}^{u} \sigma_i h(Y_i') \right]$$

$$= 2 \mathfrak{R}_u^{(\text{ind})}(\mathcal{H}). \tag{30}$$

Here ① follows from (29). ② is due to $\mathbb{E}_{\mathbf{Y}^{(u)'}} \left[ 1/u \cdot \sum_{i=1}^{u} h(Y_i') \right] = \mathcal{L}_n(h)$. ③ is due to the Jensen's inequality, and ④ is due to the definition of the Rademacher variables. All the other bounds for $\mathfrak{R}_u^-(\mathcal{H})$, $\mathfrak{R}_m^+(\mathcal{H})$, and $\mathfrak{R}_m^-(\mathcal{H})$ in (6) can be proved in a similar manner.

$\square$

## B.2. Concentration Inequality for Test-Train Process: Proof of Theorem 3.1

Let $\mathcal{H}$ be the function class defined in Section 3 of the main paper. For all $h \in \mathcal{H}$, we define

$$E(h, \mathbf{d}, \mathbf{d}^{(i)}) := \mathcal{U}_h^{(u)}(\mathbf{Z}_{\mathbf{d}}) - \mathcal{L}_h^{(m)}(\overline{\mathbf{Z}_{\mathbf{d}}}) - \mathcal{U}_h^{(u)}(\mathbf{Z}_{\mathbf{d}^{(i)}}) + \mathcal{L}_h^{(m)}(\overline{\mathbf{Z}_{\mathbf{d}^{(i)}}})$$

as the change of the test-train loss if $\mathbf{d}$ is changed to $\mathbf{d}^{(i)}$. Then we have the following lemma showing the values of $E(h, \mathbf{d}, \mathbf{d}^{(i)})$ under four specific cases. This lemma is based on the fact that there can be at most only one pair of different elements in $\{\mathbf{Z}_{\mathbf{d}}\}$ and $\{\mathbf{Z}_{\mathbf{d}^{(i)}}\}$.

**Lemma B.1.** For any $h \in \mathcal{H}$, there are four cases for the value of $E(h, \mathbf{d}, \mathbf{d}^{(i)})$ for $i \in [u]$.

Case 1: $E(h, \mathbf{d}, \mathbf{d}^{(i)}) = \left( \frac{1}{u} + \frac{1}{m} \right) (h(\mathbf{Z}_{\mathbf{d}}(i)) - h(\mathbf{Z}_{\mathbf{d}^{(i)}}(q(i))))$,
    if $d_i \neq d_i'$, $q(i) \leq u$, $p(i) > u$,

Case 2: $E(h, \mathbf{d}, \mathbf{d}^{(i)}) = \left( \frac{1}{u} + \frac{1}{m} \right) (h(\mathbf{Z}_{\mathbf{d}}(p(i))) - h(\mathbf{Z}_{\mathbf{d}^{(i)}}(i)))$,
    if $d_i \neq d_i'$, $p(i) \leq u$, $q(i) > u$,

Case 3: $E(h, \mathbf{d}, \mathbf{d}^{(i)}) = \left( \frac{1}{u} + \frac{1}{m} \right) (h(\mathbf{Z}_{\mathbf{d}}(i)) - h(\mathbf{Z}_{\mathbf{d}^{(i)}}(i)))$,
    if $d_i \neq d_i'$, $p(i) > u$, $q(i) > u$,

Case 4: $E(h, \mathbf{d}, \mathbf{d}^{(i)}) = 0$,
    if $d_i = d_i'$ or $p(i), q(i) \leq u$.

Here

$$q(i) := \min \left\{ i' \in [i+1, u] : \mathbf{Z}_{\mathbf{d}^{(i)}}(i') = i \right\}, \tag{31}$$
$$p(i) := \min \left\{ i' \in [i+1, u] : \mathbf{Z}_{\mathbf{d}}(i') = i \right\}. \tag{32}$$

In (31) and (32), we use the convention that the min over an empty set returns $+\infty$.

*Proof.* It can be checked by running Algorithm 1 that $\mathbf{Z}_{\mathbf{d}}$ and $\mathbf{Z}_{\mathbf{d}^{(i)}}$ can differ at most by one element. As a reminder, we let $\{\mathbf{Z}\}$ denote a set containing all the elements of a vector $\mathbf{Z}$ regardless of the orders of these elements in $\mathbf{Z}$.

By the definition in (31) and (32), when $p(i) \leq u$, then $\mathbf{Z}_{\mathbf{d}^{(i)}}(p(i)) = \mathbf{Z}_{\mathbf{d}}(i)$. When $q(i) \leq u$, then $\mathbf{Z}_{\mathbf{d}}(q(i)) = \mathbf{Z}_{\mathbf{d}^{(i)}}(i)$. To see this, when $p(i) \leq u$, the element $\mathbf{Z}_{\mathbf{d}}(i)$ would be picked up at a location $i' = p(i) \in (i, u]$ in $\mathbf{Z}_{\mathbf{d}^{(i)}}$. That is, $\mathbf{Z}_{\mathbf{d}^{(i)}}(p(i)) = \mathbf{Z}_{\mathbf{d}}(i)$. When $q(i) \leq u$, the element $\mathbf{Z}_{\mathbf{d}^{(i)}}(i)$ would be picked up at a location $i' = q(i) \in (i, u]$ in $\mathbf{Z}_{\mathbf{d}}$. That is, $\mathbf{Z}_{\mathbf{d}}(q(i)) = \mathbf{Z}_{\mathbf{d}^{(i)}}(i)$.

As a result, when $d_i' = d_i$, or $p(i), q(i) \leq u$, then $\{\mathbf{Z_d}\} = \{\mathbf{Z_{d^{(i)}}}\}$, and $E(g, \mathbf{d}, \mathbf{d}^{(i)}) = 0$, which proves Case 4. Otherwise, when $d_i' \neq d_i$ and only one element of $\{p(i), q(i)\}$ is not $\infty$, then the conditions of Case 2 or Case 1 hold. When the conditions of Case 2 hold, the pair of different elements in $\{\mathbf{Z_d}\}$ and $\{\mathbf{Z_{d^{(i)}}}\}$ is $\{h(\mathbf{Z_d}(p(i))), h(\mathbf{Z_{d^{(i)}}}(i))\}$. When the conditions of Case 1 hold, the pair of different elements in $\{\mathbf{Z_d}\}$ and $\{\mathbf{Z_{d^{(i)}}}\}$ is $\{h(\mathbf{Z_d}(i)), h(\mathbf{Z_{d^{(i)}}}(q(i)))\}$. It follows that both Case 2 and Case 1 hold.

If $d_i' \neq d_i$ and both $p(i)$ and $q(i)$ are $\infty$, then the only element of $\{\mathbf{Z_d}\}$ not in $\{\mathbf{Z_{d^{(i)}}}\}$ is $\mathbf{Z_d}(i)$, and the only element of $\{\mathbf{Z_{d^{(i)}}}\}$ not in $\{\mathbf{Z_d}\}$ is $\mathbf{Z_{d^{(i)}}}(i)$, so that Case 3 holds. $\qquad\square$

The proof of Theorem 3.1 needs sampling $m$ elements from the full sample $X_n$ uniformly without replacement as the training features. To this end, let $\tilde{\mathbf{d}} = \left[\tilde{d}_1, \ldots, \tilde{d}_m\right] \in \mathbb{N}^m$ be a random vector, and $\left\{\tilde{d}_i\right\}_{i=1}^m$ are $m$ independent random variables such that $\tilde{d}_i$ takes values in $[i:n]$ uniformly at random. If we invoke function RANDPERM in Algorithm 1 with input changed from $u$ to $m$, then $\mathbf{Z_{\tilde{d}}} = \text{RANDPERM}(m)$ are the first $m$ elements of a uniformly distributed permutation of $[n]$. We use $\overline{\mathbf{Z_{\tilde{d}}}} = [n] \setminus \{\mathbf{Z_{\tilde{d}}}\}$ to denote the indices not in $\{\mathbf{Z_{\tilde{d}}}\}$. Similar to $\{\mathbf{Z_d}\}$ introduced in Section 2, $\{\mathbf{Z_{\tilde{d}}}\}$ is a random set of size $m$ sampled uniformly from $[n]$ without replacement. Let $\tilde{\mathbf{d}}' = [\tilde{d}_1', \ldots, \tilde{d}_m']$ be independent copies of $\tilde{\mathbf{d}}$, and $\tilde{\mathbf{d}}^{(i)} = [\tilde{d}_1, \ldots, \tilde{d}_{i-1}, \tilde{d}_i', \tilde{d}_{i+1}, \ldots, \tilde{d}_m]$.

For all $h \in \mathcal{H}$, we define

$$\tilde{E}(h, \tilde{\mathbf{d}}, \tilde{\mathbf{d}}^{(i)}) := \mathcal{L}_h^{(m)}(\mathbf{Z_{\tilde{d}}}) - \mathcal{U}_h^{(u)}(\overline{\mathbf{Z_{\tilde{d}}}}) - \mathcal{L}_h^{(m)}(\mathbf{Z_{\tilde{d}^{(i)}}}) + \mathcal{U}_h^{(u)}(\overline{\mathbf{Z_{\tilde{d}^{(i)}}}}). \tag{33}$$

Similar to the four cases in Lemma B.1, it follows by repeating the argument in the proof of Lemma B.1 that we have the following four cases for $\tilde{E}$ as stated in the following lemma.

**Lemma B.2.** For any $h \in \mathcal{H}$, there are four cases for the value of $\tilde{E}(h, \mathbf{d}, \mathbf{d}^{(i)})$ for $i \in [m]$.

Case 1: $\tilde{E}(h, \tilde{\mathbf{d}}, \tilde{\mathbf{d}}^{(i)}) = \left(\frac{1}{u} + \frac{1}{m}\right)\left(h(\mathbf{Z_{\tilde{d}}}(i)) - h(\mathbf{Z_{\tilde{d}^{(i)}}}(\tilde{q}(i)))\right)$,
   if $\tilde{d}_i \neq \tilde{d}_i', \tilde{q}(i) \leq m, \tilde{p}(i) > m$,

Case 2: $\tilde{E}(h, \tilde{\mathbf{d}}, \tilde{\mathbf{d}}^{(i)}) = \left(\frac{1}{u} + \frac{1}{m}\right)\left(h(\mathbf{Z_{\tilde{d}}}(\tilde{p}(i))) - h(\mathbf{Z_{\tilde{d}^{(i)}}}(i))\right)$,
   if $\tilde{d}_i \neq \tilde{d}_i', \tilde{p}(i) \leq m, \tilde{q}(i) > m$,

Case 3: $\tilde{E}(h, \tilde{\mathbf{d}}, \tilde{\mathbf{d}}^{(i)}) = \left(\frac{1}{u} + \frac{1}{m}\right)\left(h(\mathbf{Z_{\tilde{d}}}(i)) - h(\mathbf{Z_{\tilde{d}^{(i)}}}(i))\right)$,
   if $\tilde{d}_i \neq \tilde{d}_i', \tilde{p}(i) > m, \tilde{q}(i) > m$,

Case 4: $\tilde{E}(h, \tilde{\mathbf{d}}, \tilde{\mathbf{d}}^{(i)}) = 0$,
   if $\tilde{d}_i = \tilde{d}_i'$ or $\tilde{p}(i), \tilde{q}(i) \leq m$,

where

$$\tilde{q}(i) := \min\left\{i' \in [i+1, m] : \mathbf{Z_{\tilde{d}^{(i)}}}(i') = i\right\}, \tag{34}$$
$$\tilde{p}(i) := \min\left\{i' \in [i+1, m] : \mathbf{Z_{\tilde{d}}}(i') = i\right\}, \tag{35}$$

and similar to (31)-(32) in Lemma B.1, we use the convention that the min over an empty set returns $+\infty$.

We need the concept of chain associated with $\mathbf{d}, \tilde{\mathbf{d}}$, the following surrogate processes, and the following lemmas, Lemma B.3-Lemma B.6, before the proof of Theorem B.7 and Theorem B.8.

A set $\{j_1, \ldots, j_Q\} \subseteq [N]$, where $j_1 < j_2 < \ldots < j_Q$, is defined to be a chain in $[N]$ associated with $\mathbf{v}$ if $j_k = \mathbf{v}(j_{k-1})$ holds for all $k \in [2 : Q]$ when $Q \geq 2$. We define the event $\Omega$ as the event that there exists a chain $\{j_1, \ldots, j_{Q'}\}$ in $[u]$ associated with $\mathbf{d}$ with $2 \leq Q' \leq u$. Similarly, let $\tilde{\Omega}$ denote the event that there is a chain $\{j_1, \ldots, j_{Q'}\}$ in $[m]$ associated with $\tilde{\mathbf{d}}$ with $2 \leq Q' \leq m$. $\Omega(Q), \tilde{\Omega}(Q)$ are defined for chains of length not less than a general $Q$ in (Yang, 2025). It also follows from (Yang, 2025) that $\Pr[\Omega] \leq u^2/m$ when $m \gg u^2$, and $\Pr\left[\tilde{\Omega}\right] \leq m^2/u$ when $u \gg m^2$.

We define a surrogate process $\bar{t}_u(\mathbf{d}, \mathcal{H}')$ for a class of functions $\mathcal{H}'$ with ranges in $[0, H']$ $(H' > 0)$ as

$$\bar{t}_u(\mathbf{d}, \mathcal{H}') := \begin{cases} t_u(\mathbf{d}, \mathcal{H}') & \mathbf{d} \notin \Omega, \\ -\sup_{h \in \mathcal{H}'} \mathcal{L}_n(h) & \mathbf{d} \in \Omega. \end{cases} \tag{36}$$

We also define a surrogate process $\bar{t}_m(\tilde{\mathbf{d}}, \mathcal{H}')$ for a class of functions $\mathcal{H}'$ with ranges in $[0, H']$ $(H' > 0)$ as

$$\bar{t}_m(\tilde{\mathbf{d}}, \mathcal{H}') := \begin{cases} t_m(\tilde{\mathbf{d}}, \mathcal{H}') & \tilde{\mathbf{d}} \notin \tilde{\Omega}, \\ -\sup_{h \in \mathcal{H}'} \mathcal{L}_n(h) & \tilde{\mathbf{d}} \in \tilde{\Omega}. \end{cases} \tag{37}$$

**Lemma B.3.** For any $h \in \mathcal{H}$, let $\mathcal{A}_{h,2} = \left\{ i \in [u] \colon E(h, \mathbf{d}, \mathbf{d}^{(i)}) \text{ satisfies Case 2 in Lemma B.1} \right\}$ and $\tilde{\mathcal{A}}_{h,2} = \left\{ i \in [m] \colon \tilde{E}(h, \tilde{\mathbf{d}}, \tilde{\mathbf{d}}^{(i)}) \text{ satisfies Case 2 in Lemma B.2} \right\}$. Then we have

$$\sum_{i \in \mathcal{A}_{h,2}} \left( h(\mathbf{Z}_{\mathbf{d}}(p(i))) \right)^2 \leq \sum_{i=1}^{n} \left( h(\mathbf{Z}_{\mathbf{d}}(i)) \right)^2, \text{ if } \mathbf{d} \notin \Omega. \tag{38}$$

$$\sum_{i \in \tilde{\mathcal{A}}_{h,2}} \left( h(\mathbf{Z}_{\tilde{\mathbf{d}}}(\tilde{p}(i))) \right)^2 \leq \sum_{i=1}^{n} \left( h(\mathbf{Z}_{\tilde{\mathbf{d}}}(i)) \right)^2, \text{ if } \tilde{\mathbf{d}} \notin \tilde{\Omega}. \tag{39}$$

*Proof.* This lemma follows from (Yang, 2025, Lemma B.6, Lemma B.11). $\qquad\square$

**Lemma B.4.** For a function class $\mathcal{H}'$, we define

$$t_u(\mathbf{d}, \mathcal{H}') := \sup_{h \in \mathcal{H}'} \left( \mathcal{U}_h^{(u)}(\mathbf{Z}_{\mathbf{d}}) - \mathcal{L}_n(h) \right), \tag{40}$$

$$t_m(\tilde{\mathbf{d}}, \mathcal{H}') := \sup_{h \in \mathcal{H}'} \left( \mathcal{L}_h^{(m)}(\mathbf{Z}_{\tilde{\mathbf{d}}}) - \mathcal{L}_n(h) \right). \tag{41}$$

Then

$$\mathbb{E}_{\mathbf{d}} \left[ t_u(\mathbf{d}, \mathcal{H}') \right] = \mathfrak{R}_u^+(\mathcal{H}'), \quad \mathbb{E}_{\tilde{\mathbf{d}}} \left[ t_m(\tilde{\mathbf{d}}, \mathcal{H}') \right] = \mathfrak{R}_m^+(\mathcal{H}'). \tag{42}$$

Moreover, for $g(\mathbf{d})$ defined in (4), we have

$$\mathbb{E}_{\mathbf{d}} \left[ g(\mathbf{d}) \right] \leq \mathfrak{R}_u^+(\mathcal{H}) + \mathfrak{R}_m^-(\mathcal{H}). \tag{43}$$

*Proof.* $\mathbb{E}_{\mathbf{d}} \left[ t_u(\mathbf{d}) \right] = \mathfrak{R}_u^+(\mathcal{H}')$ follows from the definition of Transductive Complexity in (5). We have

$$\mathbb{E}_{\tilde{\mathbf{d}}} \left[ t_m(\tilde{\mathbf{d}}, \mathcal{H}') \right] = \mathbb{E}_{\tilde{\mathbf{d}}} \left[ \sup_{h \in \mathcal{H}'} \left( \mathcal{L}_h^{(m)}(\mathbf{Z}_{\tilde{\mathbf{d}}}) - \mathcal{L}_n(h) \right) \right]$$

$$= \mathbb{E}_{\mathbf{d}} \left[ \sup_{h \in \mathcal{H}'} \left( \mathcal{L}_h^{(m)}(\overline{\mathbf{Z}_{\mathbf{d}}}) - \mathcal{L}_n(h) \right) \right] = \mathfrak{R}_m^+(\mathcal{H}').$$

where the second last equality is due to the fact that $\{\mathbf{Z}_{\tilde{\mathbf{d}}}\}$ and $\{\overline{\mathbf{Z}_{\mathbf{d}}}\}$ have the same distribution, that is, they are sets of size $m$ sampled uniformly from $[n]$ without replacement. This proves (42).

We now prove (43). We first have

$$\mathbb{E}_{\mathbf{d}} \left[ g(\mathbf{d}) \right] = \mathbb{E}_{\mathbf{d}} \left[ \sup_{h \in \mathcal{H}} \left( \mathcal{U}_h^{(u)}(\mathbf{Z}_{\mathbf{d}}) - \mathcal{L}_h^{(m)}(\overline{\mathbf{Z}_{\mathbf{d}}}) \right) \right]$$

$$= \mathbb{E}_{\mathbf{d}} \left[ \sup_{h \in \mathcal{H}} \left( \mathcal{U}_h^{(u)}(\mathbf{Z}_{\mathbf{d}}) - \mathcal{L}_n(h) + \mathcal{L}_n(h) - \mathcal{L}_h^{(m)}(\overline{\mathbf{Z}_{\mathbf{d}}}) \right) \right]$$

$$\overset{①}{\leq} \underbrace{\mathbb{E}_{\mathbf{d}} \left[ \sup_{h \in \mathcal{H}} \left( \mathcal{U}_h^{(u)}(\mathbf{Z}_{\mathbf{d}}) - \mathcal{L}_n(h) \right) \right]}_{\mathfrak{R}_u^+(\mathcal{H})} + \underbrace{\mathbb{E}_{\mathbf{d}} \left[ \sup_{h \in \mathcal{H}} \left( \mathcal{L}_n(h) - \mathcal{L}_h^{(m)}(\overline{\mathbf{Z}_{\mathbf{d}}}) \right) \right]}_{\mathfrak{R}_m^-(\mathcal{H})}$$

$$= \mathfrak{R}_u^+(\mathcal{H}) + \mathfrak{R}_m^-(\mathcal{H})$$

Here ① follows from the sub-additivity of supremum.

$\square$

**Lemma B.5** ((Yang, 2025, Lemma B.8, Lemma B.13) with $Q = 2$)**.** Let $\mathcal{H}'$ be a class of functions with ranges in $[0, H']$, and $t_u, \bar{t}_u$ are defined in (40) and (36). Suppose $\sup_{h \in \mathcal{H}'} \mathcal{L}_n(h) \leq r$ for $r > 0$. Then

$$\log \mathbb{E}_\mathbf{d} \left[\exp\left(\lambda\left(\bar{t}_u(\mathbf{d}, \mathcal{H}') - \mathbb{E}_\mathbf{d}\left[\bar{t}_u(\mathbf{d}, \mathcal{H}')\right]\right)\right)\right] \leq \frac{2H'\lambda^2\left(\mathbb{E}_\mathbf{d}\left[t_u(\mathbf{d}, \mathcal{H}')\right] + r\right)}{u - 2H'\lambda} \tag{44}$$

holds for all $\lambda \in (0, u/(2H'))$, where the surrogate process $\bar{t}_u$ is defined in (36). Similarly,

$$\log \mathbb{E}_{\tilde{\mathbf{d}}}\left[\exp\left(\lambda\left(\bar{t}_m(\tilde{\mathbf{d}}, \mathcal{H}') - \mathbb{E}_{\tilde{\mathbf{d}}}\left[\bar{t}_m(\tilde{\mathbf{d}}, \mathcal{H}')\right]\right)\right)\right] \leq \frac{2H'\lambda^2\left(\mathbb{E}_{\tilde{\mathbf{d}}}\left[t_m(\tilde{\mathbf{d}}, \mathcal{H}')\right] + r\right)}{m - 2H'\lambda} \tag{45}$$

holds for all $\lambda \in (0, m/(2H'))$, where the surrogate process $\bar{t}_m$ is defined in (37)

**Lemma B.6** (Special case of (Yang, 2025, Lemma B.7, Lemma B.12) with $Q = 2$)**.** For any $h \in \mathcal{H}$, let

$$\mathcal{A}_{h,1} = \{i \in [u] \colon \text{Case 1 defined in Lemma B.1 is satisfied}\},$$

$$\tilde{\mathcal{A}}_{h,1} = \{i \in [m] \colon \text{Case 1 defined in Lemma B.2 is satisfied}\}.$$

Then with probability at least $1 - \Pr[\Omega]$, for all $h \in \mathcal{H}$, we have

$$\mathbb{E}\left[\sum_{i=1}^u (h(\mathbf{Z}_{\mathbf{d}^{(i)}}(i)))^2 \,\middle|\, \mathbf{d}\right] \leq \frac{nu}{m} T_n(h), \mathbb{E}\left[\sum_{i \in \mathcal{A}_{h,1}} (h(\mathbf{Z}_{\mathbf{d}^{(i)}}(q(i))))^2 \,\middle|\, \mathbf{d}\right] \leq \frac{2nu}{m} T_n(h), \tag{46}$$

$$\mathbb{E}\left[\sum_{i=1}^m (h(\mathbf{Z}_{\tilde{\mathbf{d}}^{(i)}}(i)))^2 \,\middle|\, \tilde{\mathbf{d}}\right] \leq \frac{nm}{u} T_n(h), \mathbb{E}\left[\sum_{i \in \tilde{\mathcal{A}}_{h,1}} (h(\mathbf{Z}_{\tilde{\mathbf{d}}^{(i)}}(\tilde{q}(i))))^2 \,\middle|\, \tilde{\mathbf{d}}\right] \leq \frac{2nm}{u} T_n(h). \tag{47}$$

The following theorem states the concentration inequality for the supremum of the test-train process $g(\mathbf{d})$ when $m \gg u^2$.

**Theorem B.7.** Suppose $\sup_{h \in \mathcal{H}} T_n(h^2) \leq r$. If $m \gg u^2$, then for all $x > 0$, with probability at least $1 - \exp(-x) - \Pr[\Omega]$ over $\mathbf{d}$,

$$g(\mathbf{d}) \leq \mathbb{E}_\mathbf{d}[g(\mathbf{d})] + 8\sqrt{\frac{5rx}{u}} + 2\sqrt{2} \inf_{\alpha > 0}\left(\frac{\mathfrak{R}_u^+(\mathcal{H}^2)}{\alpha} + \frac{2\alpha x}{u}\right) + \frac{8H_0^2 x}{u}. \tag{48}$$

*Proof.* This theorem follows from the proof of (Yang, 2025, Theorem 5.1) with the special case that $Q = 2$. $\square$

The following theorem, Theorem B.8, states the concentration inequality for the supremum of the test-train process $g(\mathbf{d})$ when $u \gg m^2$. Many technical details in the proof of Theorem B.7 are reused in the proof of Theorem B.8. The major difference is that we consider a different supremum of empirical process, $\sup_{h \in \mathcal{H}}\left(\mathcal{U}_h^{(u)}(\overline{\mathbf{Z}_{\tilde{\mathbf{d}}}}) - \mathcal{L}_h^{(m)}(\mathbf{Z}_{\tilde{\mathbf{d}}})\right)$, instead of $g(\mathbf{d})$ as in the proof of Theorem B.7, to handle the case that $u \geq m$.

**Theorem B.8.** Suppose $\sup_{h \in \mathcal{H}} T_n(h) \leq r$. If $u \gg m^2$, then for all $x > 0$, with probability at least $1 - \exp(-x) - \Pr\left[\tilde{\Omega}\right]$ over $\mathbf{d}$,

$$g(\mathbf{d}) \leq \mathbb{E}[g(\mathbf{d})] + 8\sqrt{\frac{5rx}{m}} + 2\sqrt{2} \inf_{\alpha > 0}\left(\frac{\mathfrak{R}_m^+(\mathcal{H}^2)}{\alpha} + \frac{2\alpha x}{m}\right) + \frac{8H_0^2 x}{m}. \tag{49}$$

*Proof.* This theorem follows from the proof of (Yang, 2025, Theorem 5.2) with the special case that $Q = 2$. $\square$

**Proof of Theorem 3.1 .** (7) follows by combining the upper bound (48) in Theorem B.7 for the case that $m \gg u^2$, and the upper bound (49) for the case that $u \gg m^2$ in Theorem B.8. In particular, we note that when $Q = 2$, $\Pr[\Omega] \leq u^2/m$ when $m \gg u^2$, and $\Pr\left[\tilde{\Omega}\right] \leq m^2/u$ when $u \gg m^2$. $\square$

### B.3. Proof of Theorem 3.2

Below are the definitions and lemmas useful for the proof of Theorem 3.2.

For $r > 0$, define the function class

$$\mathcal{H}^{(r)} = \left\{ \frac{r}{w(h)} h \colon h \in \mathcal{H} \right\}, \tag{50}$$

where $w(h) := \min \left\{ r\lambda^k \colon k \geq 0, r\lambda^k \geq \tilde{T}_n(h) \right\}$ with $\lambda > 1$.

Define

$$U_r^+ := \sup_{s \in \mathcal{H}^{(r)}} \left( \mathcal{U}_s^{(u)}(\mathbf{Z_d}) - \mathcal{L}_s^{(m)}(\overline{\mathbf{Z_d}}) \right). \tag{51}$$

**Lemma B.9.** Fix $\lambda > 1$, $K > 1$, and $r > 0$. If $U_r^+ \leq \frac{r}{\lambda K}$, then

$$\mathcal{U}_h^{(u)}(\mathbf{Z_d}) \leq \mathcal{L}_h^{(m)}(\overline{\mathbf{Z_d}}) + \frac{r}{\lambda K} + \frac{\tilde{T}_n(h)}{K}, \quad \forall h \in \mathcal{H}. \tag{52}$$

*Proof.* If $\tilde{T}_n(h) \leq r$, then $w(h) = r$ and $s = \frac{r}{w(h)} h = h$. Therefore, $U_r^+ \leq \frac{r}{\lambda} \Rightarrow \mathcal{U}_s^{(u)}(\mathbf{Z_d}) - \mathcal{L}_s^{(m)}(\overline{\mathbf{Z_d}}) \leq \frac{r}{\lambda K}$ and (52) holds since $\tilde{T}_n(h) \geq 0$ for all $h \in \mathcal{H}$.

If $\tilde{T}_n(h) > r$, then $w(h) = r\lambda^k$ with $\tilde{T}_n(h) \in (r\lambda^{k-1}, r\lambda^k]$. Again, it follows from $U_r^+ \leq \frac{r}{\lambda}$ that

$$\mathcal{U}_s^{(u)}(\mathbf{Z_d}) - \mathcal{L}_s^{(m)}(\overline{\mathbf{Z_d}}) \leq \frac{r}{\lambda}, s = \frac{h}{\lambda^k},$$

and we have

$$\mathcal{U}_h^{(u)}(\mathbf{Z_d}) - \mathcal{L}_h^{(m)}(\overline{\mathbf{Z_d}}) \leq \frac{r\lambda^{k-1}}{K} \leq \frac{\tilde{T}_n(h)}{K},$$

and (52) still holds. $\qquad\square$

**Proof of Theorem 3.2.** Let $r$ be chosen such that $r \geq \max \{r_u, r_m\}$. Let $s = \frac{r}{w(h)} h \in \mathcal{H}^{(r)}$, then we have $T_n(s) \leq r$. To see this, if $\tilde{T}_n(h) \leq r$, then $w(h) = r$ and $s = h$, so $T_n(s) \leq \tilde{T}_n(h) \leq r$. Otherwise, if $\tilde{T}_n(h) > r$, then $s = \frac{h}{\lambda^k}$ where $k$ is such that $\tilde{T}_n(h) \in (r\lambda^{k-1}, r\lambda^k]$. It follows that $T_n(s) = \frac{T_n(h)}{\lambda^{2k}} \leq \frac{\tilde{T}_n(h)}{\lambda^{2k}} \leq \frac{r\lambda^k}{\lambda^{2k}} \leq r$. It follows that $T_n(s) \leq r$ for all $s \in \mathcal{H}^{(r)}$.

We first consider the case that $m \geq u$. It follows from (43) in Lemma B.4 that $\mathbb{E}_\mathbf{d}\left[ U_r^+ \right] \leq \mathfrak{R}_u^+(\mathcal{H}^{(r)}) + \mathfrak{R}_m^-(\mathcal{H}^{(r)})$. Applying (7) in Theorem 3.1 with $\alpha = 1$ to the function class $\mathcal{H}^{(r)}$, then for all $x > 0$, with probability at least $1 - e^{-x}$,

$$U_r^+ \leq \mathfrak{R}_u^+(\mathcal{H}^{(r)}) + \Theta(\mathfrak{R}_u^+(\mathcal{H}_1^{(r)})) + \mathfrak{R}_m^-(\mathcal{H}^{(r)}) + \Theta\left( \frac{x}{u} \right) + \Theta\left( \sqrt{\frac{rx}{u}} \right), \tag{53}$$

where $\mathcal{H}_1^{(r)} = \left\{ h^2 \colon h \in \mathcal{H}^{(r)} \right\}$.

Define the function class $\mathcal{H}(x, y) := \left\{ h \in \mathcal{H} \colon x \leq \tilde{T}_n(h) \leq y \right\}$. Let $T$ be the smallest integer such that $r\lambda^{T+1} \geq T_0 := \sup_{h \in \mathcal{H}} T_n(h)$. If $T_0 = \infty$, then set $T = \infty$. We have

$$\mathfrak{R}_u^+(\mathcal{H}^{(r)}) \leq \mathbb{E}\left[ \sup_{h \in \mathcal{H}(0,r)} R_{u,\mathbf{d}}^+ h \right] + \mathbb{E}\left[ \sup_{h \in \mathcal{H}(r, T_0)} \frac{r}{w(h)} R_{u,\mathbf{d}}^+ h \right]$$

$$\leq \mathbb{E}\left[ \sup_{h \in \mathcal{H}(0,r)} R_{u,\mathbf{d}}^+ h \right] + \sum_{t=0}^{T} \mathbb{E}\left[ \sup_{h \in \mathcal{H}(r\lambda^t, r\lambda^{t+1})} \frac{r}{w(h)} R_{u,\mathbf{d}}^+ h \right]$$

$$\overset{①}{\leq} \psi_u(r) + \sum_{t=0}^{T} \lambda^{-t} \psi_u(r\lambda^{t+1})$$

$$\overset{②}{\leq} \psi_u(r) \left(1 + \lambda^{1/2} \sum_{t=0}^{T} \lambda^{-t/2}\right). \tag{54}$$

Here ① is due to $w(h) \geq r\lambda^t$ and $\mathbb{E}\left[\sup_{h:\, \tilde{T}_n(h) \leq r\lambda^{t+1}} R_{u,\mathbf{d}}^+ h\right] \leq \psi_u(r\lambda^{t+1})$. ② is due to the fact that the sub-root function $\psi$ satisfies $\psi_u(\alpha r) \leq \sqrt{\alpha} \psi_u(r)$ for $\alpha > 1$.

Setting $\lambda = 4$ on the RHS of (54), we have

$$\mathfrak{R}_u^+(\mathcal{H}^{(r)}) \leq 5\psi_u(r) \leq 5\sqrt{rr_u}. \tag{55}$$

The last inequality follows from $\psi_u(r) \leq \sqrt{\frac{r}{r_u}} \psi(r_u) = \sqrt{rr_u}$ because $r \geq r_u$.

Following a similar argument,

$$\mathfrak{R}_u^+(\mathcal{H}_1^{(r)}) \leq \mathbb{E}\left[\sup_{h \in \mathcal{H}(0,r)} R_{u,\mathbf{d}}^+ h^2\right] + \sum_{t=0}^{T} \mathbb{E}\left[\sup_{h \in \mathcal{H}(r\lambda^t, r\lambda^{t+1})} \frac{r^2}{w(h)^2} R_{u,\mathbf{d}}^+ h^2\right]$$

$$\overset{③}{\leq} \psi_u(r) + \sum_{t=0}^{T} \lambda^{-2t} \psi_u(r\lambda^{t+1})$$

$$\overset{④}{\leq} \psi_u(r) \left(1 + \lambda^{1/2} \sum_{t=0}^{T} \lambda^{-3t/2}\right). \tag{56}$$

Here ① is due to $w(h) \geq r\lambda^t$ and $\mathbb{E}\left[\sup_{h:\, \tilde{T}_n(h) \leq r\lambda^{t+1}} R_{u,\mathbf{d}}^+ h^2\right] \leq \psi_u(r\lambda^{t+1})$. ② is due to the fact that the sub-root function $\psi$ satisfies $\psi_u(\alpha r) \leq \sqrt{\alpha} \psi_u(r)$ for $\alpha > 1$.

Again, setting $\lambda = 4$ on the RHS of (56), we have

$$\mathfrak{R}_u^+(\mathcal{H}_1^{(r)}) \leq \frac{23}{7} \psi_u(r) \leq \frac{23}{7} \sqrt{rr_u}. \tag{57}$$

Similar to the argument for $\mathfrak{R}_u^+(\mathcal{H}^{(r)})$, since $r \geq r_m$, we have

$$\mathfrak{R}_m^-(\mathcal{H}^{(r)}) \leq 5\sqrt{rr_m} \tag{58}$$

It follows from (53), (55), (57), and (58) that

$$U_r^+ \leq \Theta(\sqrt{rr_u}) + \Theta(\sqrt{rr_m}) + \Theta\left(\frac{x}{u}\right) + \Theta\left(\sqrt{\frac{rx}{u}}\right) := P(r). \tag{59}$$

Let $r_0$ be the largest solution to $P(r) = \frac{r}{\lambda K}$ for the fixed $K > 1$. We have

$$r_0 \leq r_1 := \lambda \left(\lambda K^2 \left(\Theta(\sqrt{r_u}) + \Theta(\sqrt{r_m}) + \Theta\left(\sqrt{\frac{x}{u}}\right)\right)^2 + \frac{\Theta(Kx)}{u}\right). \tag{60}$$

Then $r_1 \geq \max\{r_u, r_m\}$. Setting $r = r_1$ in (59), we have

$$U_{r_1}^+ \leq \frac{r_1}{\lambda K} = \lambda K \left(\Theta(\sqrt{r_u}) + \Theta(\sqrt{r_m}) + \Theta\left(\sqrt{\frac{x}{u}}\right)\right)^2 + \Theta\left(\frac{x}{u}\right).$$

It follows from Lemma B.9 that

$$\mathcal{U}_h^{(u)}(\mathbf{Z_d}) \leq \mathcal{L}_h^{(m)}(\overline{\mathbf{Z_d}}) + \frac{r_1}{\lambda K} + \frac{\tilde{T}_n(h)}{K}$$

$$= \mathcal{L}_h^{(m)}(\overline{\mathbf{Z_d}}) + \frac{\tilde{T}_n(h)}{K} + \lambda K \left( \Theta(\sqrt{r_u}) + \Theta(\sqrt{r_m}) + \Theta \left( \sqrt{\frac{x}{u}} \right) \right)^2 + \Theta \left( \frac{x}{u} \right)$$

$$\le \mathcal{L}_h^{(m)}(\overline{\mathbf{Z_d}}) + \frac{\tilde{T}_n(h)}{K} + c_0 \left( r_u + r_m \right) + \frac{c_1 x}{u}, \quad \forall h \in \mathcal{H}. \tag{61}$$

Regarding the other case that $u \ge m$, we repeat the argument above and obtain the same upper bound as (61) with $m$ and $u$ swapped:

$$\mathcal{U}_h^{(u)}(\mathbf{Z_d}) \le \mathcal{L}_h^{(m)}(\overline{\mathbf{Z_d}}) + \frac{\tilde{T}_n(h)}{K} + c_0 \left( r_u + r_m \right) + \frac{c_1 x}{m}, \quad \forall h \in \mathcal{H}. \tag{62}$$

(10) the follows from (61) and (62). $\qquad\square$

## B.4. Sharp TLC Excess Risk Bound for Generic Transductive Learning

**Proof of Theorem 3.5.** We first check that $T_n(h) \le \tilde{T}_n(h)$. For any $h \in \mathcal{H}$, by the definition of $\tilde{T}_n(h)$ in (12), for every $\varepsilon > 0$, there exist $f_1, f_2 \in \mathcal{F}$ such that $h = \ell_{f_1} - \ell_{f_2}$, and $2B\mathcal{L}_n(\ell_{f_1} - \ell_{f_n^*}) + 2B\mathcal{L}_n(\ell_{f_2} - \ell_{f_n^*}) < \tilde{T}_n(h) + \varepsilon$. Therefore,

$$\begin{aligned}
T_n(h) &= \mathcal{L}_n \left( (\ell_{f_1} - \ell_{f_2})^2 \right) \\
&\le 2T_n \left( \ell_{f_1} - \ell_{f_n^*} \right) + 2T_n \left( \ell_{f_2} - \ell_{f_n^*} \right) \\
&\le 2B\mathcal{L}_n(\ell_{f_1} - \ell_{f_n^*}) + 2B\mathcal{L}_n(\ell_{f_2} - \ell_{f_n^*}) < \tilde{T}_n(h) + \varepsilon,
\end{aligned}$$

where the first inequality follows from the Cauchy-Schwarz inequality, and the second inequality is due to Assumption 1(2). It follows that $T_n(h) \le \tilde{T}_n(h)$.

As a result, we can apply Theorem 3.2 with $\tilde{T}_n(\cdot)$ defined in this theorem. Then (15) follows from Theorem 3.2. $\qquad\square$

**Lemma B.10.** Suppose Assumption 1 holds and $m \gg u^2$ or $u \gg m^2$. Define

$$g_u^+(\mathbf{d}) := \sup_{h \in \mathcal{H}} \left( \mathcal{U}_h^{(u)}(\mathbf{Z_d}) - \mathcal{L}_n(h) \right), \tag{63}$$

$$g_m(\tilde{\mathbf{d}}) := \sup_{h \in \mathcal{H}} \left( \mathcal{L}_n(h) - \mathcal{L}_h^{(m)}(\mathbf{Z_{\tilde{d}}}) \right), \tag{64}$$

$$g_u^-(\mathbf{d}) := \sup_{h \in \mathcal{H}} \left( \mathcal{L}_n(h) - \mathcal{U}_h^{(u)}(\mathbf{Z_d}) \right). \tag{65}$$

Suppose $\sup_{h \in \mathcal{H}} T_n(h) \le r$. Then for every $x > 0$, with probability at least $1 - \exp(-x) - (\min\{m, u\})^2 / \max\{m, u\}$ over $\mathbf{d}$,

$$g_u^+(\mathbf{d}) - \mathbb{E}_{\mathbf{d}}\left[ g_u^+(\mathbf{d}) \right] \lesssim \frac{m}{n} \left( \sqrt{\frac{rx}{\min\{u, m\}}} + \inf_{\alpha > 0} \left( \frac{\mathfrak{R}_{\min\{u,m\}}^+(\mathcal{H}^2)}{\alpha} + \frac{\alpha x}{\min\{u, m\}} \right) + \frac{x}{\min\{u, m\}} \right), \tag{66}$$

where $[\cdot]_+ := \max\{\cdot, 0\}$.

Furthermore, let $\mathcal{H} = \Delta_{\mathcal{F}}^*$, $\psi_{u,m}$ be a sub-root function and $r^*$ is the fixed point of $\psi_{u,m}$. Assume that for all $r \ge r^*$,

$$\begin{aligned}
\psi_{u,m}(r) \ge \max \Bigg\{ &\mathbb{E}\left[ \sup_{h:\, h \in \Delta_{\mathcal{F}}^*, B\mathcal{L}_n(h) \le r} R_{u,\mathbf{d}}^- h \right], \mathbb{E}\left[ \sup_{h:\, h \in \Delta_{\mathcal{F}}^*, B\mathcal{L}_n(h) \le r} R_{m,\mathbf{d}}^- h \right], \\
&\mathbb{E}\left[ \sup_{h:\, h \in \Delta_{\mathcal{F}}^*, B\mathcal{L}_n(h) \le r} R_{\min\{u,m\},\mathbf{d}}^+ h^2 \right] \Bigg\},
\end{aligned} \tag{67}$$

Then for any fixed constant $K > 1$, there exists an absolute positive constant $\hat{c}_2$ depending on $K, L_0$ such that for every $x > 0$, with probability at least $1 - \exp(-x) - (\min\{m, u\})^2 / \max\{m, u\}$ over $\tilde{\mathbf{d}}$,

$$g_m(\tilde{\mathbf{d}}) \le \frac{B\mathcal{L}_n(h)}{K} + \hat{c}_2 \left( r^* + \frac{x}{\min\{u, m\}} \right). \tag{68}$$

Similarly, with probability at least $1 - \exp(-x) - (\min\{m, u\})^2 / \max\{m, u\}$ over $\mathbf{d}$,

$$g_u^-(\mathbf{d}) \leq \frac{B\mathcal{L}_n(h)}{K} + \widehat{c}_2 \left( r^* + \frac{x}{\min\{u, m\}} \right). \tag{69}$$

*Proof.* We have $g_u^+(\mathbf{d}) = \frac{m}{n} g(\mathbf{d})$, therefore,

$$g_u^+(\mathbf{d}) - \mathbb{E}_{\mathbf{d}}\left[g_u^+(\mathbf{d})\right] = \frac{m}{n}\left(g(\mathbf{d}) - \mathbb{E}_{\mathbf{d}}\left[g(\mathbf{d})\right]\right),$$

where $g$ is defined in (4). Then (66) follows from (7) in Theorem 3.1.

It can be verified that

$$g_m(\tilde{\mathbf{d}}) \stackrel{\text{dist}}{=} \frac{u}{n} \sup_{h \in \mathcal{H}} \left( \mathcal{U}_h^{(u)}(\mathbf{Z_d}) - \mathcal{L}_h^{(m)}(\overline{\mathbf{Z_d}}) \right) = \frac{u}{n} g(\mathbf{d}),$$

where the first equality is due to the fact that $\{\mathbf{Z_d}\}$ and $\overline{\{\mathbf{Z_{\tilde{d}}}\}}$ are both sets of size $u$ sampled uniformly from $[n]$ without replacement. Here $\stackrel{\text{dist}}{=}$ indicates the random variables on both sides follow the same distribution. As a result, $\mathbb{E}_{\tilde{\mathbf{d}}}\left[g_m(\tilde{\mathbf{d}})\right] = u/n \cdot \mathbb{E}_{\mathbf{d}}\left[g(\mathbf{d})\right]$ and

$$g_m(\tilde{\mathbf{d}}) - \mathbb{E}_{\tilde{\mathbf{d}}}\left[g_m(\tilde{\mathbf{d}})\right] \stackrel{\text{dist}}{=} \frac{u}{n}\left(g(\mathbf{d}) - \mathbb{E}_{\mathbf{d}}\left[g(\mathbf{d})\right]\right),$$

and it follows from (7) in Theorem 3.1 that

$$g_m(\tilde{\mathbf{d}}) - \mathbb{E}_{\tilde{\mathbf{d}}}\left[g_m(\tilde{\mathbf{d}})\right] \lesssim \sqrt{\frac{rx}{\min\{u, m\}}} + \inf_{\alpha > 0} \left( \frac{\left[\mathfrak{R}_{\min\{u,m\}}^+(\mathcal{H}^2)\right]_+}{\alpha} + \frac{\alpha x}{\min\{u, m\}} \right) + \frac{x}{\min\{u, m\}}. \tag{70}$$

We further have

$$g_u^-(\mathbf{d}) = \frac{m}{n} \sup_{h \in \mathcal{H}} \left( \mathcal{L}_h^{(m)}(\overline{\mathbf{Z_d}}) - \mathcal{U}_h^{(u)}(\mathbf{Z_d}) \right). \tag{71}$$

Taking $\left\{\vec{\mathbf{x}}_i\right\}_{i \in \overline{\mathbf{Z_d}}}$ as the training features and $\left\{\vec{\mathbf{x}}_i\right\}_{i \in \mathbf{Z_d}}$ as the test set, then we can repeat the proofs of Theorem B.7 and Theorem B.8, and obtain the following concentration inequality. For every $x > 0$, with probability at least $1 - \exp(-x) - (\min\{m, u\})^2 / \max\{m, u\}$ over $\mathbf{d}$,

$$\mathcal{L}_h^{(m)}(\overline{\mathbf{Z_d}}) - \mathcal{U}_h^{(u)}(\mathbf{Z_d}) - \mathbb{E}_{\mathbf{d}}\left[\mathcal{L}_h^{(m)}(\overline{\mathbf{Z_d}}) - \mathcal{U}_h^{(u)}(\mathbf{Z_d})\right]$$

$$\lesssim \sqrt{\frac{rx}{\min\{u, m\}}} + \inf_{\alpha > 0} \left( \frac{\mathfrak{R}_{\min\{u,m\}}^+(\mathcal{H}^2)}{\alpha} + \frac{\alpha x}{\min\{u, m\}} \right) + \frac{x}{\min\{u, m\}}. \tag{72}$$

It follows from (71) and (72) that

$$g_u^-(\mathbf{d}) - \mathbb{E}_{\mathbf{d}}\left[g_u^-(\mathbf{d})\right] \lesssim \sqrt{\frac{rx}{\min\{u, m\}}} + \inf_{\alpha > 0} \left( \frac{\left[\mathfrak{R}_{\min\{u,m\}}^+(\mathcal{H}^2)\right]_+}{\alpha} + \frac{\alpha x}{\min\{u, m\}} \right) + \frac{x}{\min\{u, m\}}. \tag{73}$$

Furthermore, we have $\mathbb{E}_{\tilde{\mathbf{d}}}\left[g_m(\tilde{\mathbf{d}})\right] = \mathfrak{R}_m^-(\mathcal{H})$ and $\mathbb{E}_{\mathbf{d}}\left[g_u^-(\mathbf{d})\right] = \mathfrak{R}_u^-(\mathcal{H})$.

For any $h \in \mathcal{H} = \Delta_{\mathcal{F}}^*$ such that $h = \ell_f - \ell_{f_n^*}$ with $f \in \mathcal{F}$, we set $\tilde{T}_n(h) = B\mathcal{L}_n(h)$. It follows from Assumption 1(2) that $T_n(h) \leq \tilde{T}_n(h)$ for all $h \in \Delta_{\mathcal{F}}^*$.

We note that $\sup_{h:\, h\in\Delta_{\mathcal{F}}^*, B\mathcal{L}_n(h)\leq r} R^+_{\min\{u,m\},\mathbf{d}}h^2 \geq 0$, $\sup_{h:\, h\in\Delta_{\mathcal{F}}^*, B\mathcal{L}_n(h)\leq r} R^-_{u,\mathbf{d}}h \geq 0$ and $\sup_{h:\, h\in\Delta_{\mathcal{F}}^*, B\mathcal{L}_n(h)\leq r} R^-_{m,\mathbf{d}}h \geq$ holds because $0 \in \Delta_{\mathcal{F}}^*$. With $\psi_{u.m}$ given in this theorem, by repeating the proof of Theorem 3.2 to (70) and (73), we have

$$g_m(\tilde{\mathbf{d}}) \leq \frac{B\mathcal{L}_n(h)}{K} + \widehat{c}_2\left(r^* + \frac{x}{\min\{u,m\}}\right),$$

$$g_u^-(\mathbf{d}) \leq \frac{B\mathcal{L}_n(h)}{K} + \widehat{c}_2\left(r^* + \frac{x}{\min\{u,m\}}\right),$$

where $K > 1$ is a fixed constant, and $\widehat{c}_2$ is a positive constant depending on $K$ and $L_0$. $\qquad\square$

**Theorem B.11.** Suppose that Assumption 1 holds and $m \gg u^2$ or $u \gg m^2$. Let $\psi_{u,m}$ be a sub-root function and $r^*$ is the fixed point of $\psi_{u,m}$. Assume that for all $r \geq r^*$,

$$\psi_{u,m}(r) \geq \max\left\{\mathbb{E}\left[\sup_{h:\, h\in\Delta_{\mathcal{F}}^*, B\mathcal{L}_n(h)\leq r} R^-_{u,\mathbf{d}}h\right], \mathbb{E}\left[\sup_{h:\, h\in\Delta_{\mathcal{F}}^*, B\mathcal{L}_n(h)\leq r} R^-_{m,\mathbf{d}}h\right],\right.$$
$$\left.\mathbb{E}\left[\sup_{h:\, h\in\Delta_{\mathcal{F}}^*, B\mathcal{L}_n(h)\leq r} R^+_{\min\{u,m\},\mathbf{d}}h^2\right]\right\}, \tag{74}$$

Then for every $x > 0$, with probability at least $1 - 2\exp(-x) - 2\left(\min\{m,u\}\right)^2/\max\{m,u\}$,

$$\mathcal{L}_n(\ell_{\widehat{f}_{\mathbf{d},u}} - \ell_{f_n^*}) \leq c_2\left(r^* + \frac{x}{u}\right), \quad \mathcal{L}_n(\ell_{\widehat{f}_{\mathbf{d},m}} - \ell_{f_n^*}) \leq c_2\left(r^* + \frac{x}{m}\right), \tag{75}$$

where $c_2$ is an absolute positive constant which depends on $B$ and $L_0$.

*Proof.* Let $\mathcal{H} = \Delta_{\mathcal{F}}^*$. It follows from (69) in Lemma B.10 that with high probability, for all $h \in \mathcal{H}$,

$$\mathcal{L}_n(h) - \mathcal{U}_h^{(u)}(\mathbf{Z_d}) \leq \frac{B\mathcal{L}_n(h)}{K} + \widehat{c}_2\left(r^* + \frac{x}{\min\{u,m\}}\right)$$

holds for a fixed constant $K > 1$, and $\widehat{c}_2$ depends on $K$ and $L_0$. We set $h = \widehat{f}_{\mathbf{d},u} - \ell_{f_n^*}$ in the above inequality, and note that $\mathcal{U}_h^{(u)}(\mathbf{Z_d}) = \mathcal{U}_{\widehat{f}_{\mathbf{d},u}}^{(u)}(\mathbf{Z_d}) - \mathcal{U}_{f_n^*}^{(u)}(\mathbf{Z_d}) \leq 0$ due to the optimality of $\widehat{f}_{\mathbf{d},u}$. Let $K > B$, then the first upper bound in (75) is proved by the above inequality with constant $c_2 = \widehat{c}_2/(1 - B/K)$.

Moreover, it follows from (68) in Lemma B.10 that with high probability, for all $h \in \mathcal{H}$,

$$\mathcal{L}_n(h) - \mathcal{L}_h^{(m)}(\overline{\mathbf{Z_d}}) \leq \frac{B\mathcal{L}_n(h)}{K} + \widehat{c}_2\left(r^* + \frac{x}{\min\{u,m\}}\right),$$

since $\{\mathbf{Z_{\tilde{d}}}\}$ and $\{\overline{\mathbf{Z_d}}\}$ follow the same distribution and they are all random sets of size $m$ sampled uniformly from $[n]$ without replacement. We set $h = \widehat{f}_{\mathbf{d},m} - \ell_{f_n^*}$ in the above inequality, and note that $\mathcal{L}_h^{(m)}(\mathbf{Z_d}) = \mathcal{L}_{\widehat{f}_{\mathbf{d},m}}^{(m)}(\overline{\mathbf{Z_d}}) - \mathcal{L}_{f_n^*}^{(m)}(\overline{\mathbf{Z_d}}) \leq 0$ due to the optimality of $\widehat{f}_{\mathbf{d},m}$. Let $K > B$, then the second upper bound in (75) is proved by the above inequality with the same constant $c_2 = \widehat{c}_2/(1 - B/K)$.

$\qquad\square$

### B.5. TLC Excess Risk Bound for Transductive Kernel Learning

Before presenting the proof of Theorem 4.1, we introduce the definition about convex and symmetric sets below, as well as Lemma B.12 which lay the foundation of the proof of Theorem 4.1.

**Definition B.1** (Convex and Symmetric Set). A set $X$ is convex if $\alpha X + (1-\alpha)X \subseteq X$ for all $\alpha \in [0,1]$. $X$ is symmetric if $-X \subseteq X$.

**Lemma B.12.** Let $\mathcal{F} = \mathcal{H}_{\mathbf{X}_n}(\mu)$. For every $r > 0$,

$$\mathbb{E}_{\mathbf{Y}^{(u)}, \boldsymbol{\sigma}} \left[ \sup_{f \in \mathcal{F}:\, T_n(f) \leq r} R^{(\text{ind})}_{\boldsymbol{\sigma}, \mathbf{Y}^{(u)}} f \right] \leq \tilde{\varphi}_u(r), \tag{76}$$

where

$$\tilde{\varphi}_u(r) := \min_{Q:\, 0 \leq Q \leq n} \left( \sqrt{\frac{rQ}{u}} + \mu \sqrt{\frac{\sum\limits_{q=Q+1}^{n} \widehat{\lambda}_q}{u}} \right). \tag{77}$$

Similarly, for every $r > 0$,

$$\mathbb{E}_{\mathbf{Y}^{(m)}, \boldsymbol{\sigma}} \left[ \sup_{f \in \mathcal{F}:\, T_n(f) \leq r} R^{(\text{ind})}_{\boldsymbol{\sigma}, \mathbf{Y}^{(m)}} f \right] \leq \tilde{\varphi}_m(r), \tag{78}$$

where

$$\tilde{\varphi}_m(r) := \min_{Q:\, 0 \leq Q \leq n} \left( \sqrt{\frac{rQ}{m}} + \mu \sqrt{\frac{\sum\limits_{q=Q+1}^{n} \widehat{\lambda}_q}{m}} \right). \tag{79}$$

*Proof.* We have

$$R^{(\text{ind})}_{\boldsymbol{\sigma}, \mathbf{Y}^{(u)}} f = \frac{1}{u} \sum_{i=1}^{u} \sigma_i f(\vec{\mathbf{x}}_{Y_i}) = \left\langle f, \frac{1}{u} \sum_{i=1}^{u} \sigma_i K(\cdot, \vec{\mathbf{x}}_{Y_i}) \right\rangle_{\mathcal{H}_K}. \tag{80}$$

Because $\left\{ \Phi^{(k)} \right\}_{k \geq 1}$ is an orthonormal basis of $\mathcal{H}_K$, for any $0 \leq Q \leq n$, we further express the first term on the RHS of (80) as

$$\left\langle f, \frac{1}{u} \sum_{i=1}^{u} \sigma_i K(\cdot, \vec{\mathbf{x}}_{Y_i}) \right\rangle_{\mathcal{H}_K} = \left\langle \sum_{q=1}^{Q} \sqrt{\widehat{\lambda}_q} \, \langle f, \Phi_q \rangle_{\mathcal{H}_K} \Phi_q, v^{(Q)}(\mathbf{Y}^{(u)}, \boldsymbol{\sigma}) \right\rangle_{\mathcal{H}_K} + \left\langle \bar{f}, \bar{v}^{(Q)}(\mathbf{Y}^{(u)}, \boldsymbol{\sigma}) \right\rangle_{\mathcal{H}_K}, \tag{81}$$

where

$$\bar{f} = f - \sum_{q=1}^{Q} \langle f, \Phi_q \rangle_{\mathcal{H}_K} \Phi_q,$$

$$v^{(Q)}(\mathbf{Y}^{(u)}, \boldsymbol{\sigma}) := \frac{1}{u} \sum_{q=1}^{Q} \frac{1}{\sqrt{\widehat{\lambda}_q}} \left\langle \sum_{i=1}^{u} \sigma_i K(\cdot, \vec{\mathbf{x}}_{Y_i}), \Phi_q \right\rangle_{\mathcal{H}_K} \Phi_q,$$

$$\bar{v}^{(Q)}(\mathbf{Y}^{(u)}, \boldsymbol{\sigma}) := \frac{1}{u} \sum_{q=Q+1}^{n} \left\langle \sum_{i=1}^{u} \sigma_i K(\cdot, \vec{\mathbf{x}}_{Y_i}), \Phi_q \right\rangle_{\mathcal{H}_K} \Phi_q.$$

Define the operator $\widehat{T}_n \colon \mathcal{H}_K \to \mathcal{H}_K$ by $\widehat{T}_n f = 1/n \cdot \sum_{i=1}^{n} K(\cdot, \vec{\mathbf{x}}_i) f(\vec{\mathbf{x}}_i)$ for any $f \in \mathcal{H}_K$. It can be verified that $\Phi_q$ is the eigenfunction of $\widehat{T}_n$ with the corresponding eigenvalue $\widehat{\lambda}_q$ for $q \in [n]$. We have

$$\left\langle \widehat{T}_n f, f \right\rangle_{\mathcal{H}_K} = \left\langle \frac{1}{n} \sum_{i=1}^{n} K(\cdot, \vec{\mathbf{x}}_i) f(\vec{\mathbf{x}}_i), f \right\rangle_{\mathcal{H}_K} = T_n(f).$$

As a result,

$$\left\|\sum_{q=1}^{Q}\sqrt{\widehat{\lambda}_q}\left\langle f,\Phi_q\right\rangle\Phi_q\right\|_{\mathcal{H}_K}^2 = \sum_{q=1}^{Q}\widehat{\lambda}_q\left\langle f,\Phi_q\right\rangle_{\mathcal{H}_K}^2 \le \sum_{q=1}^{n}\widehat{\lambda}_q\left\langle f,\Phi_q\right\rangle_{\mathcal{H}_K}^2 = \left\langle T_n f,f\right\rangle_{\mathcal{H}_K} = T_n(f) \le r, \tag{82}$$

which holds for all $f$ such that $T_n(f) \le r$.

Combining (80)-(82), we have

$$\begin{aligned}
\mathbb{E}_{\mathbf{Y}^{(u)},\boldsymbol{\sigma}}&\left[\sup_{f\in\mathcal{F}:\,T_n(f)\le r}R_{\boldsymbol{\sigma},\mathbf{Y}^{(u)}}^{(\text{ind})}f\right]\\
&\overset{\textcircled{1}}{\le}\sup_{f\in\mathcal{F}:\,T_n(f)\le r}\left\|\sum_{q=1}^{Q}\sqrt{\widehat{\lambda}_q}\left\langle f,\Phi_q\right\rangle_{\mathcal{H}_K}\Phi_q\right\|_{\mathcal{H}_K}\cdot\mathbb{E}_{\mathbf{Y}^{(u)},\boldsymbol{\sigma}}\left[\left\|v^{(Q)}(\mathbf{Y}^{(u)},\boldsymbol{\sigma})\right\|_{\mathcal{H}_K}\right]\\
&\quad+\left\|\bar{f}\right\|_{\mathcal{H}_K}\cdot\mathbb{E}_{\mathbf{Y}^{(u)},\boldsymbol{\sigma}}\left[\left\|\bar{v}^{(Q)}(\mathbf{Y}^{(u)},\boldsymbol{\sigma})\right\|_{\mathcal{H}_K}\right]\\
&\le\sqrt{r}\,\mathbb{E}_{\mathbf{Y}^{(u)},\boldsymbol{\sigma}}\left[\left\|v^{(Q)}(\mathbf{Y}^{(u)},\boldsymbol{\sigma})\right\|_{\mathcal{H}_K}\right]+\mu\mathbb{E}_{\mathbf{Y}^{(u)},\boldsymbol{\sigma}}\left[\left\|\bar{v}^{(Q)}(\mathbf{Y}^{(u)},\boldsymbol{\sigma})\right\|_{\mathcal{H}_K}\right].
\end{aligned} \tag{83}$$

where $\textcircled{1}$ is due to the Cauchy-Schwarz inequality.

We have

$$\begin{aligned}
\frac{1}{u}\mathbb{E}_{\mathbf{Y}^{(u)},\boldsymbol{\sigma}}\left[\left\langle\sum_{i=1}^{u}\sigma_i K(\cdot,\vec{\mathbf{x}}_{Y_i}),\Phi_q\right\rangle_{\mathcal{H}_K}^2\right]&\overset{\textcircled{1}}{=}\frac{1}{u}\mathbb{E}_{\mathbf{Y}^{(u)}}\left[\sum_{i=1}^{u}\left\langle K(\cdot,\vec{\mathbf{x}}_{Y_i}),\Phi_q\right\rangle_{\mathcal{H}_K}^2\right]\\
&=\frac{1}{u}\mathbb{E}_{\mathbf{Y}^{(u)}}\left[\sum_{i=1}^{u}\Phi_q(\vec{\mathbf{x}}_{Y_i})^2\right]\\
&=\frac{1}{n}\sum_{i=1}^{n}\Phi_q^2(\vec{\mathbf{x}}_i)\\
&=\left\langle\widehat{T}_n\Phi_q,\Phi_q\right\rangle=\widehat{\lambda}_q.
\end{aligned} \tag{84}$$

Here $\textcircled{1}$ is due to the fact that $\mathbb{E}[\sigma_i]=0$ for all $i\in[n]$. It follows from (84) that

$$\begin{aligned}
\mathbb{E}_{\mathbf{Y}^{(u)},\boldsymbol{\sigma}}\left[\left\|v^{(Q)}(\mathbf{Y}^{(u)},\boldsymbol{\sigma})\right\|_{\mathcal{H}_K}\right]&=\frac{1}{\sqrt{u}}\mathbb{E}_{\mathbf{Y}^{(u)},\boldsymbol{\sigma}}\left[\sqrt{\frac{1}{u}\sum_{q=1}^{Q}\frac{1}{\widehat{\lambda}_q}\left\langle\sum_{i=1}^{u}\sigma_i K(\cdot,\vec{\mathbf{x}}_{Y_i}),\Phi_q\right\rangle_{\mathcal{H}_K}^2}\right]\\
&\overset{\textcircled{1}}{\le}\frac{1}{\sqrt{u}}\sqrt{\frac{1}{u}\mathbb{E}_{\mathbf{Y}^{(u)},\boldsymbol{\sigma}}\left[\sum_{q=1}^{Q}\frac{1}{\widehat{\lambda}_q}\left\langle\sum_{i=1}^{u}\sigma_i K(\cdot,\vec{\mathbf{x}}_{Y_i}),\Phi_q\right\rangle_{\mathcal{H}_K}^2\right]}\\
&\overset{\textcircled{2}}{=}\sqrt{\frac{Q}{u}},
\end{aligned} \tag{85}$$

where $\textcircled{1}$ is due to the Jensen's inequality, $\textcircled{2}$ is due to the fact that $\mathbb{E}[\sigma_i]=0$ for all $i\in[n]$. Similarly, we have

$$\begin{aligned}
\mathbb{E}_{\mathbf{Y}^{(u)},\boldsymbol{\sigma}}\left[\left\|\bar{v}^{(Q)}(\mathbf{Y}^{(u)},\boldsymbol{\sigma})\right\|_{\mathcal{H}_K}\right]&=\frac{1}{\sqrt{u}}\mathbb{E}_{\mathbf{Y}^{(u)},\boldsymbol{\sigma}}\left[\sqrt{\frac{1}{u}\sum_{q=Q+1}^{n}\left\langle\sum_{i=1}^{u}\sigma_i K(\cdot,\vec{\mathbf{x}}_{Y_i}),\Phi_q\right\rangle_{\mathcal{H}_K}^2}\right]\\
&\le\frac{1}{\sqrt{u}}\sqrt{\frac{1}{u}\mathbb{E}_{\mathbf{Y}^{(u)},\boldsymbol{\sigma}}\left[\sum_{q=Q+1}^{n}\left\langle\sum_{i=1}^{u}\sigma_i K(\cdot,\vec{\mathbf{x}}_{Y_i}),\Phi_q\right\rangle_{\mathcal{H}_K}^2\right]}
\end{aligned}$$

$$= \sqrt{\frac{\sum\limits_{q=Q+1}^{n} \widehat{\lambda}_q}{u}}. \tag{86}$$

It follows from (83), (85), and (86) that

$$\mathbb{E}_{\mathbf{d},\boldsymbol{\sigma}} \left[ \sup_{f \in \mathcal{F} : \, T_n(f) \leq r} \left\langle f, \frac{1}{u} \sum_{i=1}^{u} \sigma_i K(\cdot, \vec{\mathbf{x}}_{Y_i}) \right\rangle \right] \leq \min_{Q : \, 0 \leq Q \leq n} \left( \sqrt{\frac{rQ}{u}} + \mu \sqrt{\frac{\sum\limits_{q=Q+1}^{n} \widehat{\lambda}_q}{u}} \right), \tag{87}$$

which completes the proof of (76). (78) can be proved by a similar argument. $\qquad\square$

The following corollary will also be necessary for the proof of Theorem 4.1. We can have $\psi_u$, $\psi_m$ in Theorem 3.5 as the upper bounds for inductive Rademacher complexities using Theorem 2.1, leading to this corollary.

**Corollary B.13.** Under the same conditions of Theorem 3.5, for every $x > 0$, with probability at least $1 - \exp(-x) - (\min\{m, u\})^2 / \max\{m, u\}$, (15) still holds if for all $r \geq r_u$,

$$\psi_u(r) \geq 2 \max \left\{ \mathbb{E} \left[ \sup_{h : \, h \in \Delta_{\mathcal{F}}, \tilde{T}_n(h) \leq r} R_{\boldsymbol{\sigma}, \mathbf{Y}^{(u)}}^{(\text{ind})} h \right], \mathbb{E} \left[ \sup_{h : \, h \in \Delta_{\mathcal{F}}, \tilde{T}_n(h) \leq r} R_{\boldsymbol{\sigma}, \mathbf{Y}^{(u)}}^{(\text{ind})} h^2 \right] \right\}, \tag{88}$$

and for all $r \geq r_m$,

$$\psi_m(r) \geq 2 \max \left\{ \mathbb{E} \left[ \sup_{h : \, h \in \Delta_{\mathcal{F}}, \tilde{T}_n(h) \leq r} R_{\boldsymbol{\sigma}, \mathbf{Y}^{(m)}}^{(\text{ind})} h \right], \mathbb{E} \left[ \sup_{h : \, h \in \Delta_{\mathcal{F}}, \tilde{T}_n(h) \leq r} R_{\boldsymbol{\sigma}, \mathbf{Y}^{(m)}}^{(\text{ind})} h^2 \right] \right\}. \tag{89}$$

Here $r_u, r_m$ are the fixed points of $\psi_u$ and $\psi_m$ respectively.

Instead of proving Theorem 4.1, we will prove the more detailed version of Theorem 4.1 as shown in the following theorem.

**Theorem B.14.** Suppose that Assumption 1 (1) and Assumption 2 hold. Suppose $K$ is a positive definite kernel on $\mathcal{X} \times \mathcal{X}$. Suppose that for all $f \in \mathcal{H}_{\mathbf{X}_n}(\mu)$, $0 \leq \ell_f(i) \leq L_0$ for all $i \in [n]$, and $L_0 \geq 2\sqrt{2}$. Suppose that $m \gg u^2$ or $u \gg m^2$. Then for every $x > 0$, with probability at least $1 - \exp(-x) - (\min\{m, u\})^2 / \max\{m, u\}$ over $\mathbf{d}$,

$$\mathcal{U}_h^{(u)}(\mathbf{Z_d}) \leq \mathcal{L}_h^{(m)}(\overline{\mathbf{Z_d}}) + \frac{2L^2 B'}{K} \left( \mathcal{L}_n(\ell_{f_1} - \ell_{f_n^*}) + \mathcal{L}_n(\ell_{f_2} - \ell_{f_n^*}) \right)$$
$$+ c_3 \min_{0 \leq Q \leq n} r(u, m, Q) + \frac{c_1 x}{\min\{m, u\}}, \forall h \in \Delta_{\mathcal{H}_{\mathbf{X}_n}(\mu)}, \tag{90}$$

where $c_3$ is an absolute positive number depending on $B', L_0, L, \mu$, and

$$r(u, m, Q) := Q \left( \frac{1}{u} + \frac{1}{m} \right) + \left( \sqrt{\frac{\sum\limits_{q=Q+1}^{n} \widehat{\lambda}_q}{u}} + \sqrt{\frac{\sum\limits_{q=Q+1}^{n} \widehat{\lambda}_q}{m}} \right).$$

In particular, with probability at least $1 - 3 \exp(-x) - 3 (\min\{m, u\})^2 / \max\{m, u\}$ over $\mathbf{d}$, with $h = \widehat{f}_{\mathbf{d},m} - \widehat{f}_{\mathbf{d},u}$ in (90), we have the excess risk bound

$$\mathcal{E}(\widehat{f}_{\mathbf{d},m}) \leq c_5 \left( \min_{0 \leq Q \leq n} r(u, m, Q) + \frac{x}{\min\{m, u\}} \right), \tag{91}$$

where $c_5$ is an absolute positive constant depending on $B', L_0, L, \mu$.

**Proof of Theorem B.14.** It follows from Assumption 2 that for all $h \in \Delta_{\mathcal{F}}^*$, $T_n(h) \leq B'L^2\mathcal{L}_n(h)$. To see this, let $h = \ell_{f_1} - \ell_{f_n^*}$ with $f_1, f_2 \in \mathcal{F}$. Then $T_n(h) = T_n(\ell_{f_1} - \ell_{f_n^*}) \leq L^2 T_n(f_1 - f_n^*) \leq B'L^2\mathcal{L}_n(\ell_{f_1} - \ell_{f_n^*}) = B'L^2\mathcal{L}_n(h)$. This inequality indicates that Assumption 1 (2) holds with $B = B'L^2$. As a result, Assumption 1 holds.

We now apply Theorem 3.5 and Corollary B.13 with the function class $\mathcal{F} = \mathcal{H}_{\mathbf{X}_n}(\mu)$ and $\tilde{T}_n(\cdot)$ defined in (12) with $B = B'L^2$. Let $h = \ell_{f_1} - \ell_{f_2} \in \Delta_{\mathcal{F}}$ with $f_1, f_2 \in \mathcal{F}$, and $\tilde{T}_n(h) \leq r$. By the definition of $\tilde{T}_n$, there exist $f_1, f_2 \in \mathcal{F}$ such that $h = \ell_{f_1} - \ell_{f_2}$ and $2B\mathcal{L}_n(\ell_{f_1} - \ell_{f_n^*}) + 2B\mathcal{L}_n(\ell_{f_2} - \ell_{f_n^*}) \leq r'$ for arbitrary $r' > r$. For simplicity of notations we set $r' = 1.1r$. Let $\boldsymbol{\sigma} = \{\sigma_i\}_{i=1}^{\max\{u,m\}}$ be iid Rademacher variables. For $r > 0$ we have

$$
\begin{aligned}
& 2\mathbb{E}_{\mathbf{d}}\left[\sup_{h:\, h\in\Delta_{\mathcal{F}}, \tilde{T}_n(h)\leq r} R_{\boldsymbol{\sigma},\mathbf{Y}^{(u)}}^{(\mathrm{ind})}h\right] \\
& \overset{\text{①}}{\leq} 2\mathbb{E}_{\mathbf{Y}^{(u)},\boldsymbol{\sigma}}\left[\sup_{f_1,f_2\in\mathcal{F}:\, 2B\mathcal{L}_n(\ell_{f_1}-\ell_{f_n^*})+2B\mathcal{L}_n(\ell_{f_2}-\ell_{f_n^*})\leq r'} R_{\boldsymbol{\sigma},\mathbf{Y}^{(u)}}^{(\mathrm{ind})}(\ell_{f_1}-\ell_{f_2})\right] \\
& \leq 2\mathbb{E}_{\mathbf{Y}^{(u)},\boldsymbol{\sigma}}\left[\sup_{f_1\in\mathcal{F}:\, \mathcal{L}_n(\ell_{f_1}-\ell_{f_n^*})\leq 1.1r/2B} R_{\boldsymbol{\sigma},\mathbf{Y}^{(u)}}^{(\mathrm{ind})}\left(\ell_{f_1}-\ell_{f_n^*}\right)\right] \\
& \quad + 2\mathbb{E}_{\mathbf{Y}^{(u)},\boldsymbol{\sigma}}\left[\sup_{f_2\in\mathcal{F}:\, \mathcal{L}_n(\ell_{f_2}-\ell_{f_n^*})\leq 1.1r/2B} R_{\boldsymbol{\sigma},\mathbf{Y}^{(u)}}^{(\mathrm{ind})}\left(\ell_{f_2}-\ell_{f_n^*}\right)\right] \\
& \overset{\text{②}}{\leq} 4L\mathbb{E}_{\mathbf{Y}^{(u)},\boldsymbol{\sigma}}\left[\sup_{f\in\mathcal{F}:\, T_n(f-f_n^*)\leq rB_1/2B} R_{\boldsymbol{\sigma},\mathbf{Y}^{(u)}}^{(\mathrm{ind})}\left(f-f_n^*\right)\right] \\
& \overset{\text{③}}{\leq} 8L\mathbb{E}_{\mathbf{Y}^{(u)},\boldsymbol{\sigma}}\left[\sup_{f\in\mathcal{F}:\, T_n(f)\leq rB_1/8B} R_{\boldsymbol{\sigma},\mathbf{Y}^{(u)}}^{(\mathrm{ind})}f\right] \\
& \overset{\text{④}}{\leq} 8L\tilde{\varphi}_u\left(\frac{rB_1}{8B}\right).
\end{aligned}
\tag{92}
$$

Here ① is due to the definition of $\tilde{T}_n$. ② is due to the contraction property in Theorem A.5 and the fact that the loss function $\ell(\cdot,\cdot)$ is $L$-Lipschitz continuous, and $B_1$ is a positive constant such that $B_1 = 1.1B'$. ③ follows by noting that $(f - f_n^*)/2 \in \mathcal{F}$ because $\mathcal{F}$ is symmetric and convex. $\tilde{\varphi}_u$ in ④ is defined in (77) in Lemma B.12.

It follows from (92) that

$$
2\mathbb{E}_{\mathbf{Y}^{(u)},\boldsymbol{\sigma}}\left[\sup_{h:\, h\in\Delta_{\mathcal{F}}, \tilde{T}_n(h)\leq r} R_{\boldsymbol{\sigma},\mathbf{Y}^{(u)}}^{(\mathrm{ind})}h\right] \leq 8L\tilde{\varphi}_u\left(\frac{rB_1}{8B}\right).
$$

By a similar argument and noting that $(\ell_{f_1} - \ell_{f_2})^2 \leq 2\left((\ell_{f_1} - \ell_{f_n^*})^2 + (\ell_{f_2} - \ell_{f_n^*})^2\right)$, we have

$$
\begin{aligned}
& 2\mathbb{E}_{\mathbf{Y}^{(u)},\boldsymbol{\sigma}}\left[\sup_{h:\, h\in\Delta_{\mathcal{F}}, \tilde{T}_n(h)\leq r} R_{\boldsymbol{\sigma},\mathbf{Y}^{(u)}}^{(\mathrm{ind})}h^2\right] \\
& \leq 8\mathbb{E}_{\mathbf{Y}^{(u)},\boldsymbol{\sigma}}\left[\sup_{f\in\mathcal{F}:\, T_n(f-f^*)\leq rB_1/2B} R_{\boldsymbol{\sigma},\mathbf{Y}^{(u)}}^{(\mathrm{ind})}\left(\ell_{f_1}-\ell_{f_n^*}\right)^2\right] \\
& \overset{\text{①}}{\leq} 16L_0\mathbb{E}_{\mathbf{Y}^{(u)},\boldsymbol{\sigma}}\left[\sup_{f\in\mathcal{F}:\, T_n(f-f^*)\leq rB_1/2B} R_{\boldsymbol{\sigma},\mathbf{Y}^{(u)}}^{(\mathrm{ind})}\left(\ell_{f_1}-\ell_{f_n^*}\right)\right] \\
& \leq 32L_0L\tilde{\varphi}_u\left(\frac{rB_1}{8B}\right),
\end{aligned}
\tag{93}
$$

where ① is due to the contraction property in Theorem A.5 and $0 \leq \ell_f(i) \leq L_0$ for all $f \in \mathcal{F}$ and $i \in [n]$. Define $\varphi_u(r) := \max\left\{8L\tilde{\varphi}_u\left(\frac{rB_1}{8B}\right), 32L_0L\tilde{\varphi}_u\left(\frac{rB_1}{8B}\right)\right\} = L'\tilde{\varphi}_u\left(\frac{rB_1}{8B}\right)$ with $L' := \max\{8L, 32L_0L\}$. It can be verified that $\varphi_u$ is a sub-root function by checking the definition of the sub-root function.

Similarly, we have

$$2\mathbb{E}_{\mathbf{Y}^{(m)},\boldsymbol{\sigma}}\left[\sup_{h\colon h\in\Delta_{\mathcal{F}},\tilde{T}_n(h)\leq r}R^{(\text{ind})}_{\boldsymbol{\sigma},\mathbf{Y}^{(m)}}h\right]\leq 8L\tilde{\varphi}_m\left(\frac{rB_1}{8B}\right),$$

$$2\mathbb{E}_{\mathbf{Y}^{(m)},\boldsymbol{\sigma}}\left[\sup_{h\colon h\in\Delta_{\mathcal{F}},\tilde{T}_n(h)\leq r}R^{(\text{ind})}_{\boldsymbol{\sigma},\mathbf{Y}^{(m)}}h^2\right]\leq 32L_0L\tilde{\varphi}_m\left(\frac{rB_1}{8B}\right),$$

and $\varphi_m(r):=\max\left\{8L\tilde{\varphi}_m\left(\frac{rB_1}{8B}\right),32L_0L\tilde{\varphi}_m\left(\frac{rB_1}{8B}\right)\right\}=L'\tilde{\varphi}_m\left(\frac{rB_1}{8B}\right)$ is also a sub-root function. Let $r_u,r_m$ be the fixed point of $\varphi_u$ and $\varphi_m$ respectively. We define $\varphi(r):=\varphi_u(r)+\varphi_m(r)$, then $\varphi$ is also a sub-root function. Let $r$ be the fixed point of $\varphi$, then $r\geq\max\{r_u,r_m\}$. Since both $\varphi_u$ and $\varphi_m$ are nondecreasing functions, we have

$$r=\varphi(r)=\varphi_u(r)+\varphi_m(r)\geq\varphi_u(r_u)+\varphi_m(r_m)=r_u+r_m.$$

It then follows from the above inequality and Corollary B.13 that, for all $h\in\Delta_{\mathcal{F}}$ we have

$$\mathcal{U}^{(u)}_h(\mathbf{Z_d})\leq\mathcal{L}^{(m)}_h(\overline{\mathbf{Z_d}})+\frac{2B}{K}\left(\mathcal{L}_n(\ell_{f_1}-\ell_{f_n^*})+\mathcal{L}_n(\ell_{f_2}-\ell_{f_n^*})\right)+c_0r+\frac{c_1x}{\min\{m,u\}}.\tag{94}$$

Let $0\leq r'\leq r$ . Then it follows from (Bartlett et al., 2005, Lemma 3.2) that $0\leq r'\leq\varphi(r')$. Therefore, by the definition of $\tilde{\varphi}_u$ in (77) and $\tilde{\varphi}_m$ in (79), for every $0\leq Q\leq n$ we have

$$\frac{r'}{L'}\leq\sqrt{\frac{r'B_1Q}{8Bu}}+\sqrt{\frac{r'B_1Q}{8Bm}}+\mu\sqrt{\frac{\sum\limits_{q=Q+1}^{n}\widehat{\lambda}_q}{u}}+\mu\sqrt{\frac{\sum\limits_{q=Q+1}^{n}\widehat{\lambda}_q}{m}}.$$

Solving the above quadratic inequality for $r'$, we have

$$r'\leq\widehat{c}_3Q\left(\frac{1}{u}+\frac{1}{m}\right)+\widehat{c}_3\left(\sqrt{\frac{\sum\limits_{q=Q+1}^{n}\widehat{\lambda}_q}{u}}+\sqrt{\frac{\sum\limits_{q=Q+1}^{n}\widehat{\lambda}_q}{m}}\right)=\widehat{c}_3r(u,m,Q),\tag{95}$$

where $\widehat{c}_3$ is a positive constants depending on $B',L_0,L,\mu$. (95) holds for every $0\leq Q\leq n$, so it follows from (94) and (95) that

$$\mathcal{U}^{(u)}_h(\mathbf{Z_d})\leq\mathcal{L}^{(m)}_h(\overline{\mathbf{Z_d}})+\frac{2B}{K}\left(\mathcal{L}_n(\ell_{f_1}-\ell_{f_n^*})+\mathcal{L}_n(\ell_{f_2}-\ell_{f_n^*})\right)$$

$$+c_0\widehat{c}_3\min_{0\leq Q\leq n}\left(Q\left(\frac{1}{u}+\frac{1}{m}\right)+\left(\sqrt{\frac{\sum\limits_{q=Q+1}^{n}\widehat{\lambda}_q}{u}}+\sqrt{\frac{\sum\limits_{q=Q+1}^{n}\widehat{\lambda}_q}{m}}\right)\right)+\frac{c_1x}{\min\{m,u\}},\tag{96}$$

which proves (90) with $c_3=c_0\widehat{c}_3$.

When $h=\ell_{\widehat{f}_{\mathbf{d},m}}-\ell_{\widehat{f}_{\mathbf{d},u}}$, then we can set $f_1=\widehat{f}_{\mathbf{d},m}$ and $f_2=\widehat{f}_{\mathbf{d},u}$ in (96).

We now derive the upper bounds for $\mathcal{L}_n(\ell_{\widehat{f}_{\mathbf{d},u}}-\ell_{f_n^*})$ and $\mathcal{L}_n(\ell_{\widehat{f}_{\mathbf{d},m}}-\ell_{f_n^*})$ using Theorem B.11. Applying Theorem 2.1, we need to find the sub-root function $\psi_{u,m}$ such that

$$\psi_{u,m}(r)\geq 2\max\left\{\mathbb{E}\left[\sup_{h\colon h\in\Delta_{\mathcal{F}}^*,B\mathcal{L}_n(h)\leq r}R^{(\text{ind})}_{\boldsymbol{\sigma},\mathbf{Y}^{(u)}}h\right],\mathbb{E}\left[\sup_{h\colon h\in\Delta_{\mathcal{F}}^*,B\mathcal{L}_n(h)\leq r}R^{(\text{ind})}_{\boldsymbol{\sigma},\mathbf{Y}^{(m)}}h\right],\right.$$

$$\left.\mathbb{E}\left[\sup_{h\colon h\in\Delta_{\mathcal{F}}^*,B\mathcal{L}_n(h)\leq r}R^{(\text{ind})}_{\boldsymbol{\sigma},\mathbf{Y}^{(\min\{u,m\})}}h^2\right]\right\},$$

By repeating the argument in (92) and (93), we have

$$\psi_{u,m}(r) = \Theta\left(\min_{0\leq Q\leq n}\left(\sqrt{\frac{rQ}{u}} + \sqrt{\frac{rQ}{m}} + \mu\sqrt{\frac{\sum\limits_{q=Q+1}^{n}\widehat{\lambda}_q}{u}} + \mu\sqrt{\frac{\sum\limits_{q=Q+1}^{n}\widehat{\lambda}_q}{m}}\right)\right).$$

Let $r^*$ be the fixed point of $\psi_{u,m}$. Any $r' \leq r^*$ satisfies $r' \leq \Theta\left(\min_{0\leq Q\leq n} r(u,m,Q)\right)$. As a result, it follows from (75) in Theorem B.11 that, with probability at least $1 - 2\exp(-x) - 2\left(\min\{m,u\}\right)^2 / \max\{m,u\}$,

$$\mathcal{L}_n(\ell_{\widehat{f}_{\mathbf{d},u}} - \ell_{f_n^*}) \leq c_2\left(\Theta\left(\min_{0\leq Q\leq n} r(u,m,Q)\right) + \frac{x}{u}\right), \mathcal{L}_n(\ell_{\widehat{f}_{\mathbf{d},m}} - \ell_{f_n^*}) \leq c_2\left(\Theta\left(\min_{0\leq Q\leq n} r(u,m,Q)\right) + \frac{x}{m}\right). \tag{97}$$

We note that $\mathcal{L}^{(m)}_{\ell_{\widehat{f}_{\mathbf{d},m}} - \ell_{\widehat{f}_{\mathbf{d},u}}}(\overline{\mathbf{Z}_{\mathbf{d}}}) \leq 0$ due to the optimality of $\widehat{f}_{\mathbf{d},m}$. Applying the upper bound in (97) to (96) proves (91).

$\square$

