# OpenReview forum: "A New Concentration Inequality for Sampling Without Replacement and Its Application for Transductive Learning"
_ICML.cc/2025/Conference — ICML 2025 poster_

### Official Review · Reviewer_etXN · 2025-02-18

**Overall Recommendation:** 3

**Summary:**

The paper studies transductive learning where the training examples and testing examples are drawn from a given dataset without replacement. The problem builds generalization guarantees for transductive learning based on local Rademacher complexities. To this aim, the paper first develops a new concentration inequality for sampling without replacement, based on which the paper gives generalization bounds as well as excess risk bounds. The main contribution is to develop rates fast rates under Bernstein-type conditions. Applications to kernel learning are also given.

## After rebuttal

Thank you for your reply to my comments. It would be beneficial if the authors can include more relevant discussions with related work on concentration of sampling without replacement.

**Claims And Evidence:**

The claim is supported by theoretical analysis such as concentration inequalities for sampling without replacement, generalization bounds and excess risk bounds.

**Essential References Not Discussed:**

As far as I see, the paper gives discussions with all the essential references.

**Experimental Designs Or Analyses:**

This is a theoretical paper and there are no experimental results.

**Methods And Evaluation Criteria:**

The paper uses several methods such as concentration inequalities, local Rademacher complexity and peeling arguments. These methods are appropriate to develop fast rates for transductive learning.

**Other Comments Or Suggestions:**

- abstract: "are use"
- abstract: "we have As an application"
- Eq (1) is too small
- line 84: ". under the same assumptions"
- Theorem A.6: there should be an assumption that $f$ is convex.
- Thm 3.6: what is the meaning of $r^*$?
- line 501: "to develop Let"
- Section 6: "As an result"
- Eq (32): the meaning of $V_+$ is not given
- Eq (27): the meaning of $\psi$ is not given
- line 790: "Here (1) follow"
- line 1065: :inequality follow:

**Other Strengths And Weaknesses:**

**Strength**
- The paper gives new concentration inequality which only considers the deviation between the error in the training dataset and test dataset, without considering the error in test dataset. This seems to be better than the deviation considered in [R1] as it does not introduce $n/u$ and $n/m$, where $m$ is the size of training dataset, $u$ is the size of testing dataset and $n=u+m$. This is interesting if either $u$ or $m$ is very small.
- The derived generalization bounds and excess risk bounds are stronger than the existing results if either $u$ or $m$ is small.

**Weakness**
- The paper is not well written, and involves many notations which may be hard to digest for readers. There are also many typos.
- The proof is technical and not easy to follow. For example, in Section B.2, the paper introduces $E[h,d,d^{(i)}]$. However, the paper does not give explanations on the motivation and how it is used in the proof.
- The result only shows advantage when $m$ and $\mu$ are imbalanced. For the typical setting with $m$ and $\mu$ of the same order, the paper does not give improvements.
- The paper only gives applications to traditional transductive kernel learning problems. The paper does not consider more practical applications such as graph neural networks.

**Questions For Authors:**

Is it possible to give applications to graph neural networks, which are natural examples of transductive learning?

**Relation To Broader Scientific Literature:**

The paper is an extension of [R1], which also considered transductive learning based on local Rademacher complexity. The paper made a clear comparison with [R1], showing the advantage of the result in some special case. In particular, the theoretical result is stronger when the number of training examples and the number of testing examples are imbalanced. The paper gives examples to show that the excess risk bound in this paper always converges to 0, while the bound in [R1] can diverge when the ratio of training dataset size is too large or too small.


[R1] Tolstikhin, Ilya, Gilles Blanchard, and Marius Kloft. "Localized complexities for transductive learning." Conference on Learning Theory. PMLR, 2014.

**Theoretical Claims:**

I only checked The proof of Theorem 2.1, which is correct. I did not check other parts of the proof.

---

> ### Author Rebuttal · Authors · 2025-04-01
>
> We appreciate the review and the suggestions in this review. The raised issues are addressed below.
>
> **(1) “…many notations which may be hard to digest for readers. There are also many typos.”**
>
> Thank you for your suggestions. We have proofread this paper and fixed all the typos. In particular, all the typos in your “Other Comments Or Suggestions” have been fixed.
>
> **(2) “The proof is technical and not easy to follow. For example, in Section B.2, the paper introduces…**
>
> Following your suggestion, we will add more explanations and motivations to give readers a better understanding of the notations and their roles in our proofs. In particular, $E(h,\mathbf d, \mathbf d^{(i)})$ is the change of the test-train loss if only the $i$-th element of $\mathbf d$ is changed (that is, $\mathbf d$ is cbanged to $\mathbf d^{(i)}$). The reason we introduced $E(h,\mathbf d, \mathbf d^{(i)})$ is that it is needed in the computation of the upper variance $V_+(Z)$, and such upper variance $V_+(Z)$ will be used in the exponential version of the Efron-Stein inequality (Eq. (23)) to derive the upper bound of the logarithm of the moment generating function of $Z-E[Z]$. Such upper bound of the moment generating function of $Z-E[Z]$ will in turn be used to derive the test-train bound for transductive learning in the proof of Theorem 3.1.
>
> Furthermore, please also kindly refer to part (1) of our response to Reviewer PUBb for a detailed roadmap of our proofs.
>
> **(3) “The result only shows advantage when m and u are imbalanced. For the typical setting with m and u of the same order…”**
>
> The key advantage of our transductive learning bounds in this paper over those in [Tolstikhin et al., 2014] is that our bounds are sharper than those in [Tolstikhin et al., 2014] for unbalanced training/test features. It is important to note that our bounds, such as the excess risk bound for transductive kernel learning in Theorem 4.1, address both unbalanced and balanced training/test features in a unified framework.
>
> **(4)”…The paper does not consider more practical applications such as graph neural networks.”**
>
> Following the suggestion of this review, we will add another application to linear Graph Neural Networks (GNNs) for a transductive node classification task in the final version of this paper.
>
> **Addressing Other Comments Or Suggestions**
>
> We will fix all the typos and reformat Eq. (1). $r^*$ in Theorem 3.6 is the $r^*$ in Theorem B.11, which will be defined before Theorem 3.6 in the final version of this paper. We will add the assumption that $f$ is convex in Theorem A.6.
>
> $V_+(Z)$ defined in Theorem A.2 is the “upper variance” of the random variable $Z$ as a function of $n$ independent random variables $X_1,\ldots, X_n$. In particular,  $V_+(Z)$ measures the variance of $Z$ when $X_i$ is changed to another sample $X'_i$ for all $i \in [n]$.
>
> $\psi$ is defined as $\psi(x) = x(e^x-1)$ which will be added to Proposition A.4.
>
>
>
>  **References**
>
> [Tolstikhin et al., 2014] Tolstikhin, I. O., Blanchard, G., and Kloft, M. Localized complexities for transductive learning. COLT 2014.

---

> > ### Comment · Reviewer_etXN · 2025-04-07
> >
> > Thank you for your reply to my comments. I also suggest the authors include more relevant discussions with related work on concentration of sampling without replacement. Please note that all the references of the paper are published before 2014. The paper should include more recent discussions to show that it would receive attention of the ML community.

---

> > > ### Author Response · Authors · 2025-04-08
> > >
> > > Thank you for your suggestion! In the final version of this paper, we will discuss more related works about concentration inequalities about sampling without replacement, including [Bardenet et al., 2015, Tolstikhin2017, Sambale et al., 2021].  Such discussion will compare our transductive learning bound in Theorem 5.1 of Section 5 of this paper to [Tolstikhin2017] and reveal that our bound is still sharper using a similar argument in Section 5. We will also mention that in contrast with our results, the supremum of empirical process involving sampling without replacement is not addressed in [Bardenet et al., 2015]. We will also discuss the specific results in [Sambale et al., 2021] about the concentration inequalities on the multislice which are based on the modified log-Sobolev inequalities.
> > >
> > > **References**
> > >
> > > [Bardenet et al., 2015] R. Bardenet, O. Maillard. Concentration inequalities for sampling without replacement. Bernoulli 2015.
> > >
> > > [Tolstikhin2017] I.O. Tolstikhin. Concentration Inequalities for Samples without Replacement. Theory of Probability & Its Applications 2017.
> > >
> > > [Sambale et al., 2021] H. Sambale, A. Sinulis. Concentration Inequalities on the Multislice and for Sampling Without Replacement.  Journal of Theoretical Probability 2021.

---

### Official Review · Reviewer_Dn1f · 2025-03-10

**Overall Recommendation:** 2

**Summary:**

This paper studies the generalization of transductive learning. To do so, this paper proves a new concentration inequality for the test-train process, which is used to derive a sharp concentration inequality for the general supremum of empirical process involving random variables in the setting of sampling uniformly without replacemenn. Then, this paper uses Transductive Local
Complexity (TLC) along with the concentration inequality to provide an excess risk analysis for the empirical risk minimizer of Transductive learning.

**Claims And Evidence:**

Yes

**Essential References Not Discussed:**

NA

**Experimental Designs Or Analyses:**

NA

**Methods And Evaluation Criteria:**

NA

**Other Comments Or Suggestions:**

Although this paper studies an important problem, there is much I do not understand. I will revise my score based on the author's response.

**Other Strengths And Weaknesses:**

Strengths:

This paper is written well.

Transductive learning is important in machine learning. Better bounds are provided, including sharper concentration inequalities for sampling without replacement and sharper generalization bounds for Transductive learning.

Weaknesses:

There are too many symbols which makes this paper hard to read.

What is the difference between the strategy of sampling without replacement and Algorithm 1?

Is the bound in Theorem 3.5 sharp? For the term \frac{x}{\min\{m,u\}},  would \min\{m,u\} in the denominator make it suboptimal?

Why the Rademacher complexity is also defined on h^2. This is different from the case of inductive learning.

Why the authors claim that their concentration inequalities and generalization bounds are sharp? There are no lower bounds given.

**Questions For Authors:**

See the above

**Relation To Broader Scientific Literature:**

NA

**Theoretical Claims:**

I have checked part of the proof.

---

> ### Author Rebuttal · Authors · 2025-04-01
>
> We appreciate the review and the suggestions in this review. The raised issues are addressed below.
>
> **(1) Minimax Optimal Lower Bound for Transductive Kernel Learning**
>
> We consider the following setup for Transductive Kernel Learning (TKL) wherein we will derive the minimax optimal lower bound for this setup with a matching upper bound by Theorem 3.5. **Such results would show that Theorem 3.5 gives sharp upper bound**.
>
> Transductive Setting: Suppose that the full-sample $\mathbf X_n = (\mathbf x_i, i \in [n])$ are i.i.d. samples distributed uniformly on the unit sphere $\mathcal S^{d-1}$ in $\mathbb R^d$.  This is one standard transductive setting considered in [Tolstikhin et al., 2016] where transductive lower bound is given.
>
> Suppose that kernel $K$ is defined on $\mathcal S^{d-1}$  such that the eigenvalue of the integral operator associated with $K$ is $\lambda_j \asymp j^{-2\alpha}$ with $\alpha > 1/2$. For example, $2\alpha = d/(d-1)$ holds for the arccosine kernel. We consider the function class $\mathcal F = \mathcal H_{\mathbf X_m} \cap \mathcal H_K(\mu)$ with a positive constant $\mu$.  We assume that the target function $f^* \in \mathcal H_K(\mu)$ and $y_i = f^*(\mathbf x_i) + w_i$ where $w_i$ for $i \in [n]$ are i.i.d. Gaussian noise with mean $0$ and variance $\sigma^2$. $\mathcal U_f$ and $\mathcal L_f$ are the test loss and the training loss defined as $\mathcal U_f  = \sum_{\mathbf x \in \mathbf X_u} (f(\mathbf x)-f^*(\mathbf x))^2$ and $\mathcal L_f  = \sum_{\mathbf x \in \mathbf X_m} (f(\mathbf x)-f^*(\mathbf x))^2$.
>
> We have the following minimax lower bound. “With high probability” means that the probability goes to $1$ with $m \to \infty$.
>
> **Theorem (Minimax Lower Bound)**. Suppose $u \ge \Theta(m^{(2d+1)/(2d-1)})$, then with high probability, $\inf_{f \in \mathcal F}\sup_{f^* \in \mathcal H_K(\mu)}\mathcal U_f  \ge \Theta(m^{-d/(2d-1)})$.
> Proof. First of all, let $E(f-f^*)^2$ be the $L^2$-distance between $f$ and $f^*$ in the $L^2$ space. Then we have $\inf_{f \in \mathcal F } \sup_{f^* \in \mathcal H_K(\mu)}  E(f-f^*)^2  \ge \Theta(m^{-d/(2d-1)})$ by repeating the proof of the standard minimax lower bound in [Wainwright 2019] (e.g. Example 15.23). Then for every $f \in \mathcal F$ there exists $\hat f^*(f) \in \mathcal H_K(\mu)$ such that
> $E(f- \hat f^*(f))^2  \ge \Theta(m^{-d/(2d-1)})$. We apply the standard concentration inequality so that
> $\mathcal U_{f} - E(f-\hat f^*(f))^2   \ge -\Theta(1/{\sqrt u})$ holds for all $f, \hat f^*(f)$ with high probability.
> With $u \ge \Theta(m^{(2d+1)/(2d-1)})$, we have $\mathcal U_{f} \ge -\Theta(m^{-(d+0.5)/(2d-1)}) + E(f- \hat f^*(f))^2 \ge \Theta(m^{-d/(2d-1)}) $ holds for all $f$ and $\hat f^*(f)$, completing the proof.
>
> Based on our Theorem 3.5, we have the matching upper bound below. Suppose $g_{\mathbf \beta(T)}$ is the kernel regressor obtained by performing $T$ steps of gradient descent on the training objective
> $L(\mathbf \beta) = \sum_{\mathbf x \in \mathbf X_m} (g_{\mathbf \beta}(\mathbf x) – y(\mathbf x)  )^2$ where $g_{\mathbf \beta}(\cdot) = \sum_{\mathbf x \in \mathbf X_m} K(\cdot,\mathbf x) \mathbf \beta(\mathbf x)$. For a training point $\mathbf x$, $\beta(\mathbf x)$ and $y(\mathbf x)$ are the coefficient and the label of this point.
>
> **Theorem (Matching Upper Bound)**. Suppose $T \asymp \Theta(m^{d/(2d-1)}) $ and $g = g_{\mathbf \beta(T)}$. Then with high probability, $\mathcal U_{g} \le \Theta(m^{-d/(2d-1)})$.
>
> Proof. By the proof of part (a) of [Raskutti et al., 2014] we have the training loss which satisfies $\mathcal L_{g} \le \Theta(m^{-d/(2d-1)})$. By repeating the proof of Theorem 4.1, the bound for $\mathcal L_{g}$, and applying Theorem 3.5, we obtain
> $\mathcal U_{g} \le \Theta(m^{-d/(2d-1)})$.
>
> **(2) Addressing Other Issues**
>
> We will explain the notations with their motivations.  Algorithm 1 generates the test features $\mathbf X_u$ with indices in $\mathbf Z_{\mathbf d}$ sampled without replacement. $h^2$ appears in our Transductive Complexity (TC) because we used the exponential version of the Efron-Stein inequality to derive our transductive bounds which involve the square of loss functions. We note that using the bounds for TC in Theorem 2.1, we achieve the transductive bounds sharper than the current state-of-the-art in [Tolstikhin et al., 2014] and the minimax optimal bound in part (1) of this response.
>
> **References**
>
> [Tolstikhin et al., 2014] Tolstikhin, I. O., Blanchard, G., and Kloft, M. Localized complexities for transductive learning. COLT 2014.
>
> [Tolstikhin et al., 2016] Tolstikhin, I. Oi, Lopez-Paz, D. Minimax Lower Bounds for Realizable Transductive Classification. arXiv:1602.03027, 2016.
>
> [Raskutti et al., 2014] Raskutti, G., Wainwright, M. J., and Yu, B. Early stopping and non-parametric regression: an optimal data dependent stopping rule. JMLR 2014.
>
> [Wainwright 2019] M. J. Wainwright, High-Dimensional Statistics: A Non-Asymptotic Viewpoint. Cambridge University Press, 2019.

---

> > ### Comment · Reviewer_Dn1f · 2025-04-04
> >
> > According to the author's reply, I'm sorry that I still don't understand why the concentration inequality is tight. To me, a tight concentration inequality should be optimal in constants.
> >
> > I still feel Rademacher's dependence on h^2 strange, which may be due to Efron-stein's proof technique, but more naturally it should be a dependence on h similar to the inductive learning case.

---

> > > ### Author Response · Authors · 2025-04-04
> > >
> > > Thank you for your further comments, your further issues are addressed as follows.
> > >
> > > **(1) Tight/Sharp Upper Bound by Theorem 3.5 Matching Minimax Lower Bound with Difference in Only A Constant Factor**
> > >
> > > In our first response for the transudative kernel learning task, we provided and proved the following **minimax bower bound**:
> > >
> > > Suppose $u \ge \Theta(m^{(2d+1)/(2d-1)})$, then with high probability, $\inf_{f \in F}\sup_{f^* \in \mathcal H_K(\mu)}\mathcal U_f  \ge \Theta(m^{-d/(2d-1)})$.
> > >
> > > We also provided and proved the following **matching upper bound using Theorem 3.5 of this paper**:
> > >
> > > Suppose $T \asymp \Theta(m^{d/(2d-1)}) $ is the stopping time, and $g = g_{\mathbf \beta(T)}$. Then with high probability, $\mathcal U_{g} \le \Theta(m^{-d/(2d-1)})$.
> > > Here $g$ is a kernel regressor trained with $T$ steps of gradient descent.
> > >
> > > **As a result, the upper bound by Theorem 3.5 matches the minimax lower bound, and both bounds are both $ \Theta(m^{-d/(2d-1)})$ and they only differ by a constant factor. In the statistical learning literature, our upper bound is commonly regarded as a sharp bound matching the minimax lower bound.  We would respectfully point out that the broad statistical learning literature regards upper bounds matching minimax lower bounds with difference in only constant factors as sharp upper bounds, with abundant examples as even standard textbook examples summarized in [Wainwright 2019]**. To name a few, in [Wainwright 2019], Example 13.14 gives a sharp upper bound for nonparametric regression considered the best bound that can be obtained, with a matching minimax lower bound in Example 15.23, and the upper bound and the matching minimax lower bound have difference in only a constant factor.
> > >
> > > **(2) Main Results about Transductive Bounds (Theorem 3.5-3.6) without $h^2$**
> > >
> > > Herein we provide different versions of our main transudative bounds in Theorem 3.5-3.6 without $h^2$.
> > >
> > > First, it follows from Theorem 2.1 of this paper that the Transductive Complexity (TC) is bounded by its inductive Rademacher complexity counterpart, so that the term in Theorem 3.5 involving $h^2$, which is $E_{\sup_{h \colon h \in \Delta_{\mathcal F},\tilde T_n(h) \le r} R^{+}_{u,\mathbf d} h^2}$, can be bounded from above by
> > >
> > > $ E_{\sup_{h \colon h \in \Delta_{\mathcal F},\tilde T_n(h) \le r} R^{(\textup{ind})}_{\mathbf \sigma,\mathbf Y^{(u)}}} h^2 $,
> > >
> > > which is further upper bounded by
> > >
> > > $2L_0 E_{\sup_{h \colon h \in \Delta_{\mathcal F},\tilde T_n(h) \le r} R^{(\textup{ind})}_{\mathbf \sigma,\mathbf Y^{(u)}}} h $,
> > > due to the contraction property of the inductive Rademacher complexity.  Here $L_0 \ge 2\sqrt 2$ is the upper bound for the loss function introduced in line 266 of this paper. In this way, we can have the following corollary from Theorem 3.5 without $h^2$.
> > >
> > > **Corollary of Theorem 3.5 Without $h^2$.**  Suppose that $\psi_u$ is a sub-root function with fixed point $r_u$, and $\psi_m$ is another sub-root function with fixed point $r_m$. Assume that for all $r \ge r_u$,
> > > $$\psi_u(r) \ge 2 \max(L_0,1) E_{\sup_{h \colon h \in \Delta_{\mathcal F},\tilde T_n(h) \le r} R^{(\textup{ind})}_{\mathbf \sigma,\mathbf Y^{(u)}}} h,$$
> > > and for all $r \ge r_m$,
> > >
> > > $$\psi_m(r) \ge 2 \max(L_0,1) E_{\sup_{h \colon h \in \Delta_{\mathcal F},\tilde T_n(h) \le r} R^{(\textup{ind})}_{\mathbf \sigma,\mathbf Y^{(m)}}} h.$$
> > >
> > > Then the upper bound (15) in Theorem 3.5 still holds with the same probability and the constants in  Theorem 3.5.
> > >
> > > Similarly, we can have the following corollary from Theorem 3.6 without $h^2$.
> > >
> > > **Corollary of Theorem 3.6 Without $h^2$.**  Suppose that $\psi_{u,m}$ is a sub-root function with fixed point $r^*$, Assume that for all $r \ge r^*$,  $\psi_{u,m}(r)$ is an upper bound for $2E_{\sup_{h \colon h \in \Delta^*_{\mathcal F},B L_n(h) \le r} R^{(\textup{ind})}_{\mathbf \sigma,\mathbf Y^{(u)}}} h$,
> > >
> > > $2E_{\sup_{h \colon h \in \Delta^*_{\mathcal F},B L_n(h) \le r} R^{(\textup{ind})}_{\mathbf \sigma,\mathbf Y^{(m)}}} h $, and
> > >
> > > $2L_0 E_{\sup_{h \colon h \in \Delta^*_{\mathcal F},B L_n(h) \le r} R^{(\textup{ind})}_{\mathbf \sigma,\mathbf Y^{(\min(u,m))}}} h $.
> > >
> > > Then the upper bound (16) in Theorem 3.6 still holds with the same probability and the constants in  Theorem 3.6.
> > >
> > > We also remark that using the above results without $h^2$ does not affect the sharpness. The excess risk bound for the transductive kernel learning task in Theorem 4.1 is in fact derived following the above strategy, with TC bounded by its inductive Rademacher complexity counterpart, which leads to the excess risk bound shaper than the current state-of-the-art [Tolstikhin et al., 2014].
> > >
> > > Please kindly let us know if you have remaining questions, and we appreciate your time.
> > >
> > >
> > > **References**
> > >
> > > [Wainwright 2019] M. J. Wainwright, High-Dimensional Statistics: A Non-Asymptotic Viewpoint. Cambridge University Press, 2019.
> > >
> > > [Tolstikhin et al., 2014] Tolstikhin, I. O., Blanchard, G., and Kloft, M. Localized complexities for transductive learning. COLT 2014.

---

### Official Review · Reviewer_PUBb · 2025-03-12

**Overall Recommendation:** 4

**Summary:**

The paper presented an improvement of concentration inequality for sampling without replacement. More particular, the improvement is compared to the work of Tolstikhin et al,, 2014, in which, its upper bound of excess risk shown does not have the ratio between  test and train data size over the whole given data size.  Section 4 gives similar results for transductive kernel learning and section 5 contains results for sampling uniformly without replacement of empirical process. It is purely theoretical paper.

**Claims And Evidence:**

Since the proofs are put in the appendix and without a paragraph presenting/summarizing/comparing the techniques of this work over the work of Tolstikhin et al,, 2014 and popular works in concentration inequalities, I find it is difficult to follow the argument and the flow of the paper over 25 pages of the Appendix. This might be due to my lack of experience on transductive learning, although I am familiar with most types of concentration inequalities. Despite of these, I think the proofs are well-ordered and appear to be neat. Although the authors point out some key steps in the proofs, but for me it is difficult to connect the dots.

**Essential References Not Discussed:**

Important related works are Tolstikhin et al, 2014 and El-Yaniv & Pechyony, 2009, and some concentration inequalities. Techniques in those works are essential to understand this work.

**Experimental Designs Or Analyses:**

No experiments or analyses.

**Methods And Evaluation Criteria:**

New method of proof is presented.

**Other Comments Or Suggestions:**

No

**Other Strengths And Weaknesses:**

Strength: the new approach to tackle the concentration inequality in a particular setting.

Weakness: no presented application.

**Questions For Authors:**

For all concentration inequalities presented in the paper, the LHS is  an empirical loss, the RSH is an expected loss plus some terms. One of the key term is $r$, which is upper bound for the loss of the train and test data. So $r$ is related to the LHS as well. At this point, I find the inequality is not in "usual" form, since we want to bound the LHS and the LHS "appears again" in the bound.

**Relation To Broader Scientific Literature:**

This work is mostly related to the work of Tolstikhin et al., 2014. I am not aware/ do not know  of further/broader application except ones in the paper.

**Theoretical Claims:**

I could not check all the proofs, I did read some part of the Appendix,  I believe they are correct.

---

> ### Author Rebuttal · Authors · 2025-04-01
>
> We appreciate the review and the suggestions in this review. The raised issues are addressed below.
>
> **(1)  Proof Roadmap**
>
> Following the suggestion, we will give a detailed proof roadmap in the final main paper.  In particular, we first obtain a novel concentration inequality of the test-train process in Theorem 3.1. Applying the peeling strategy to Theorem 3.1, we obtain the first transductive learning bound in Theorem 3.2 which involves the surrogate variance operator $\tilde T_n$ and the Transductive Local Complexity (TLC). By taking a particular form of the surrogate variance operator in Eq. (12), we obtain the refined transductive learning bound in Theorem 3.5 under the standard assumption (Assumption 1) with bounded loss functions.  The excess risk bound in Theorem 3.6 is then a consequence of Theorem 3.5 and the auxiliary results in Theorem B.11. The excess risk bound for transductive kernel learning in Theorem 4.1 is obtained by applying Theorem 3.5 and Theorem 3.6 to the transductive kernel learning task.  We will also provide an illustration of the above roadmap in the final version of this paper for better understanding of our proof roadmap and proof strategies.
>
> **(2) Applications of the Transductive Learning Bounds in This Paper**
>
> We would like to mention that the transductive learning bounds in this paper have been applied to transductive kernel learning with the excess risk bound for transductive kernel learning in Theorem 4.1. We will add another application to linear Graph Neural Networks (GNNs) for a transductive node classification task in the final version of this paper.
>
> **(3) “…, since we want to bound the LHS and the LHS "appears again" in the bound.”**
>
> We respectfully point out that the LHS of our transductive bound is usually the test loss, and the RHS is usually the sum of the training loss and $r$ which is the fixed point of some sub-root function. We agree that such $r$ involves both the training features and the test features, but this does not suggest that the LHS to be bounced also appears on the RHS. In particular, the LHS is the test loss computed on the test features and the labels on the test features, while $r$ is computed on the training features and the test features (without the labels on the test features). In this sense, we respectfully disagree that LHS "appears again" in the RHS (the bound). We also remark that the learner for a transductive learning task can access both the training and the test features. For an example, in our excess risk bound for the transductive kernel learning in Theorem 4.1, $r = r(u,m,Q)$ is defined in terms of the eigenvalues of the gram matrix $\mathbf K$ over the full sample including the training features and the test features. Such $r(u,m,Q)$ can in fact be estimated from $\mathbf K$ as the transductive learning algorithm has access to the full sample. Even though both the LHS (the test loss) and the RHS ($r(u,m,Q)$) depend on the test features, $r(u,m,Q)$ does not involve the labels on the test features, so the derived bound for the LHS is still reasonable and sharper than the current state-of-the-art in [Tolstikhin et al., 2014].
>
>  **References**
>
> [Tolstikhin et al., 2014] Tolstikhin, I. O., Blanchard, G., and Kloft, M. Localized complexities for transductive learning. COLT 2014.

---

### Official Review · Reviewer_sB2h · 2025-03-14

**Overall Recommendation:** 4

**Summary:**

The main contribution of this paper is presenting a new tool, Transductive Local Complexity (TLC), to analyze the generalization of transductive learning algorithms. In the way, the authors proved a new concentration inequality with certain advantages that may be of independent interest. They also showed a few applications of their new tool and concentration inequality.

**Claims And Evidence:**

I verified the correctness of the proofs of theorems in section 3. Moreover, the applications make sense to me.

**Essential References Not Discussed:**

I think you need to add a related work section to your paper. For instance, see the subsection Background in the following paper: https://arxiv.org/abs/2405.05190.

**Ethical Review Concerns:**

-

**Experimental Designs Or Analyses:**

This is a theoretical paper without any experiment.

**Methods And Evaluation Criteria:**

The evaluation criteria are natural.

**Other Comments Or Suggestions:**

I suggest adding another application. Maybe something related to neural networks?

**Other Strengths And Weaknesses:**

This is a solid theoretical paper. I think it is a good contribution to the field of statistical learning theory.

**Questions For Authors:**

I do not have any specific question.

**Relation To Broader Scientific Literature:**

Transitive learning is a fundamental topic in learning theory. Any good result in this context will be valuable. This paper provides a new tool for analyzing the generalization of transductive learning algorithms.

**Theoretical Claims:**

I verified the correctness of the proofs of theorems in section 3.

---

> ### Author Rebuttal · Authors · 2025-04-01
>
> We appreciate the positive review and the suggestions in this review.  We will add a "Related Work" section to the final version of this paper following your suggestion. We will also discuss the application of our transudative learning bounds to linear Graph Neural Networks (GNNs) for a transudative node classification task in the final version of paper.

---

### Decision · Program_Chairs · 2025-05-01

**Decision:**

Accept (poster)

**Comment:**

The main contribution of this paper is presenting a new tool, Transductive Local Complexity (TLC), to analyze the generalization of transductive learning algorithms. In the way, the authors proved a new concentration inequality with certain advantages that may be of independent interest. They also showed a few applications of their new tool and concentration inequality.